# Near-surface softening and healing in eastern Honshu associated with the 2011 magnitude-9 Tohoku-Oki Earthquake

Su-Yang Wang [1,6], Hai-Yang Zhuang[2,6], Hao Zhang[1], Hong-Jun He[1], Wei-Ping Jiang[3], Er-Lei Yao[4], Bin Ruan[1], Yong-Xin Wu[5] & Yu Miao [1✉]

The near-surface part of the crust, also called the skin of the earth, is the arena of human activity of which the stiffness is of great concern to engineers in infrastructure construction. The stiffness reduction of near-surface geomaterials also plays a vital role in geohazards triggering. However, the physical mechanism behind the material softening is still not fully understood. Here, we report a coseismic shear-wave velocity reduction in the near surface by up to a few tens of percent during the strongest shaking from the 11 March 2011 Tohoku-Oki Earthquake and a subsequent two-stage healing process including a rapid recovery within a few minutes and a slow recovery over many years. We also present a theoretical contact model between mineral grains in geomaterials containing multiple metastable contacts at small separations due to the oscillatory hydration interaction, which can explain the emergence of different stages in the healing process.

[1] School of Civil and Hydraulic Engineering, Huazhong University of Science and Technology, Wuhan, China. [2] Institute of Geotechnical Engineering, Nanjing Tech University, Nanjing, China. [3] Institute of Engineering Mechanics, China Earthquake Administration, Harbin, China. [4] Changjiang River Scientific Research Institute, Wuhan, China. [5] Hohai University, Nanjing, China. [6] These authors contributed equally: Su-Yang Wang, Hai-Yang Zhuang. ✉email: miaoyu@hust.edu.cn

It has been known that strong ground motion produced by large earthquakes can cause softening of the Earth's near surface, even leading to liquefaction in extreme cases[1,2]. In recent decades, this phenomenon has aroused wide concern among researchers in different fields. Since the late 1980s, seismologists and geotechnical engineers have shown great interest in nonlinear amplification of ground motion by near-surface sediments in large earthquakes[3]. They found that due to the decrease in shear modulus and the increase in hysteretic damping, the ground-motion amplification at high frequencies is reduced during strong ground motions, and the resonant frequencies of local site move toward lower frequencies[4–6]. Meanwhile, physicists and geophysicists have focused their attention on nonequilibrium dynamical behavior of granular geomaterials such as soils and rocks[7]. It was recognized that granular geomaterials exhibit reversible mechanical nonlinearity depending on wave amplitude at very small deformations, which can be well understood in terms of quasi-equilibrium thermodynamics (Landau's theory). However, a wide variety of geomaterials display complex nonlinear features beyond a certain bulk strain of $\sim 10^{-7}$–$10^{-6}$ [8]. For example, geomaterials show memory softening at relatively large deformations, meaning that the modulus slowly returns to initial equilibrium state over a long period after being disturbed. Moreover, in such case, the degree of material softening is not only related to wave amplitude, but also depends on wave duration[9].

Numerous studies have reported a sudden material softening in the upper crust caused by a large earthquake, and a subsequent healing that varies logarithmically with time by monitoring seismic velocity changes from ambient noise or earthquake records[10–14]. Both static strain induced by crustal deformation and dynamic strain resulting from strong ground motion are considered to be the main controlling factors[15–17]. However, in the shallow subsurface, increasingly evidences suggest that the material softening is dominated by strong ground motions because the dynamic strain is generally much higher than the static strain within a depth of a few hundred meters[18–20]. Hence, from a physical point of view, near-surface softening and healing are determined by the dynamic elastic properties of geomaterials in the shallow subsurface, which are commonly characterized by (1) strong nonlinear elasticity, resulting in the degradation of elastic modulus; (2) stress–strain hysteresis, related to the attenuation of seismic waves; and (3) slow dynamics, also known as slow relaxation or aging behavior, linked with the long-term healing process.

The 11 March 2011 Tohoku-Oki Earthquake was the first well-recorded Mw9-class event in history, originating from the subduction of the Pacific plate beneath a continental plate at a rate of about 9 cm/year (Fig. 1). This earthquake ruptured a megathrust fault along the Japan Trench offshore of northeastern Honshu, with a large coseismic slip exceeding 30 m near the epicenter[21]. On the eastern coast of northern Honshu, the closest land region to the epicenter, the maximum horizontal displacement of crustal deformation reached up to 5.3 m[22,23] (corresponding to a static strain of $10^{-5}$–$10^{-4}$ [24]), and the highest peak acceleration of strong ground motion exceeded 20 m/s² [25](corresponding to a dynamic strain of $10^{-3}$–$10^{-2}$ [18,26]). Applying deconvolution or cross-correlation-based interferometry to vertical array records, researchers have measured a sharp reduction in the near-surface shear-wave velocity during the strongest ground motion of the main shock by up to 30%[11,27] and a decreasing velocity reduction by about 5–10% in the first month after the main shock[18,26] followed by a logarithmic healing over time in the next several months throughout northeastern Honshu[11,18]. However, the coseismic and postseismic changes in near-surface shear-wave velocity associated with the 2011 Tohoku-Oki earthquake are still not fully understood. The major problems remaining to be solved

are: (1) when the near-surface softening ends; (2) how the recovery rate changes over time; (3) what is the physical mechanism underlying the slow dynamics.

In this study, we aim to investigate the near-surface softening and healing associated with the 2011 Mw9.0 Tohoku-Oki Earthquake by performing deconvolution analyses with the seismic records of more than 3700 earthquakes at 12 KiK-net stations between June 2000 and March 2019 (see the section "Methods"). We first analyzed the coseismic and postseismic velocity variations from seismological field observations. Then we compared the velocity recovery processes obtained from field observations and laboratory experiments. Finally, we developed a basic theoretical model describing the slow recovery, and examined the model with experimental data obtained in laboratory and from the field.

## Results

**Field observations and comparisons with experiments.** We applied short-time moving-window seismic interferometry to the seismograms recorded during the Tohoku-Oki Earthquake at 12 selected KiK-net stations in eastern Honshu to reveal the coseismic changes in near-surface seismic velocity (Fig. 2). The seismic velocity began to decrease after the S-wave arrived (at 30–50 s) and continued to drop as the shaking became stronger, and then reached the lowest value during the strongest shaking (at 100–140 s). Because of a complex rupture process with a source duration that lasted for more than 120 s[21], the envelope of the seismograms during the strongest shaking may include multiple distinct peaks, especially for the seven stations in Fukushima starting with "FKSH". For these stations, the lowest seismic velocity does not necessarily correspond to the highest peak, but may correspond to a lower peak. The maximum coseismic velocity reduction was up to 46.3% at FKSH14 and ranged from 11.9% to 38.9% for other stations. Based on the results of material softening in soils and rocks observed in various laboratory tests, geotechnical engineers and physicists have reached a consensus that the degradation of modulus (or elastic wave velocity) is mainly affected by confining pressure, strain amplitude, and geomaterial type[28,29]. In a recent study, it was found that the mean effective confining pressure is likely to be similar across different KiK-net stations, hence the degree of the near-surface velocity reduction is primarily affected by the shear strain and soil type, which was empirically estimated using the peak ground acceleration, the initial unperturbed near-surface shear-wave velocity, and the plastic index of the near-surface sediments[30]. For example, according to the one-dimensional wave propagation theory in Beresnev and Wen[31], a 5-Hz sinusoidal wave with an amplitude of 1 m/s² leads to a dynamic shear strain of $1 \times 10^{-4}$ at a typical stiff-soil site of a shear-wave velocity of 320 m/s, which generates a shear modulus reduction of $\sim$5–30% depending on soil plasticity based on the data from geotechnical laboratory experiments in Vucetic and Dobry[32].

As shown in Fig. 2, for the majority of the 12 stations with the exception of FKSH09, FKSH14, FKSH17, and FKSH19, the near-surface healing process started right after the seismic velocity reached its lowest value, while the start time of the healing process was picked manually for the four exceptional stations. Specifically, with an exception of an anomalous case at FKSH14, the start time corresponds to the last significant acceleration peak during the strongest shaking for three of the four stations. The coseismic velocity variations at FKSH14 indicate suspected transient liquefaction in the shallow surface layer, which can be confirmed by two existing liquefaction detection methods using ground motion records from Kostadinov and Yamazaki[33] and is also visually evidenced by the abrupt drop in the waveform

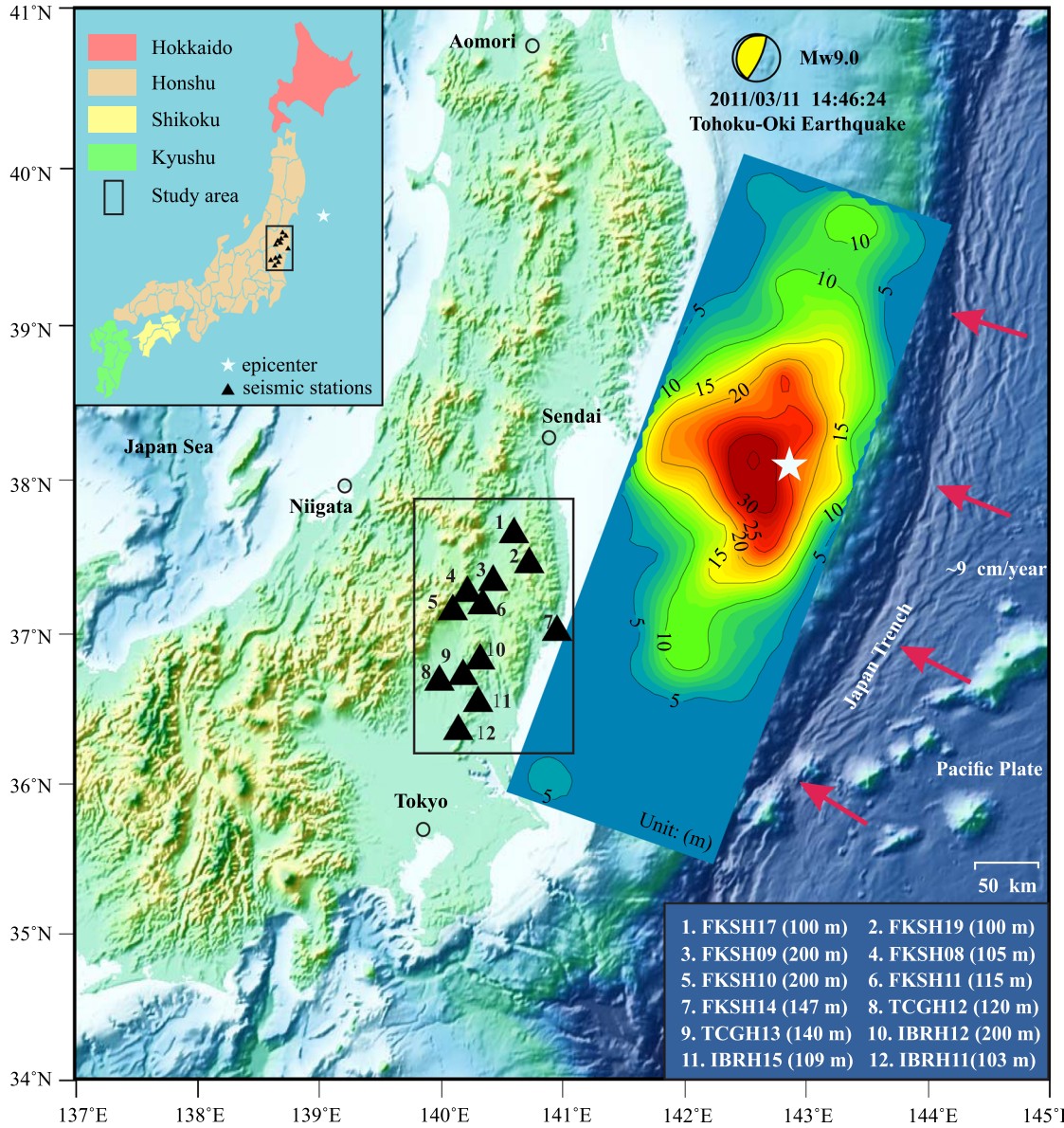

**Fig. 1 Topographic map of northern Honshu Island and surrounding area, with slip distribution of the Tohoku-Oki Earthquake[21].** The solid black triangles represent the seismic stations selected in this study. The station codes and borehole depths are listed in the bottom right corner. The white pentagram indicates the epicenter of the Tohoku-Oki Earthquake.

amplitude before and after the last significant acceleration peak of the surface seismograms (Supplementary Fig. 1). Hence, considering a solidification process after the suspected transient liquefaction, we defined the start time of the healing process as the time with the fastest velocity recovery rate for FKSH14, and this definition is also applicable to almost all the stations.

In order to further investigate the postseismic changes in near-surface shear-wave velocity associated with the Tohoku-Oki Earthquake, we also applied short-time moving-window seismic interferometry to the seismograms recorded within $10^5$ s (about 28 h) of the main shock, and then applied non-moving-window seismic interferometry to the remaining seismograms for the sake of computational efficiency. The shear-wave velocity measured at those selected stations recovered logarithmically with time in two stages after the near-surface softening caused by the Tohoku-Oki Earthquake (Fig. 3a). The first stage is a rapid recovery within 72–468 s, and the second stage is a slow recovery over up to more than 8 years ($10^{8.40}$ s). As of March 2019, the recovery of the

shear-wave velocity did not seem to have stabilized, even for three of the stations for which the shear-wave velocities had recovered to their pre-seismic level. For those stations whose near-surface seismic velocities had not yet recovered to the pre-seismic level, the time required to return to the pre-seismic level at the current recovery rate of the second stage ranged from several months to two hundreds years with a median of 15 years. We also noticed that the recovery rate in the first 10 s of the first stage at station IBRH11 was much lower than expected. We speculate that this may be related to the cyclic mobility in near-surface sediments caused by soil skeleton dilation at large shear strains during incipient quasi-liquefaction[34], which can be supported by the abrupt drop in the waveform amplitude of the surface seismograms (Supplementary Fig. 2), and it also should be noted that IBRH11 is one of the stations with the largest relative velocity changes.

The postseismic velocity variations at different stations were normalized by dividing by the initial velocity reductions at the corresponding start time (Fig. 3b). A general pattern is found in

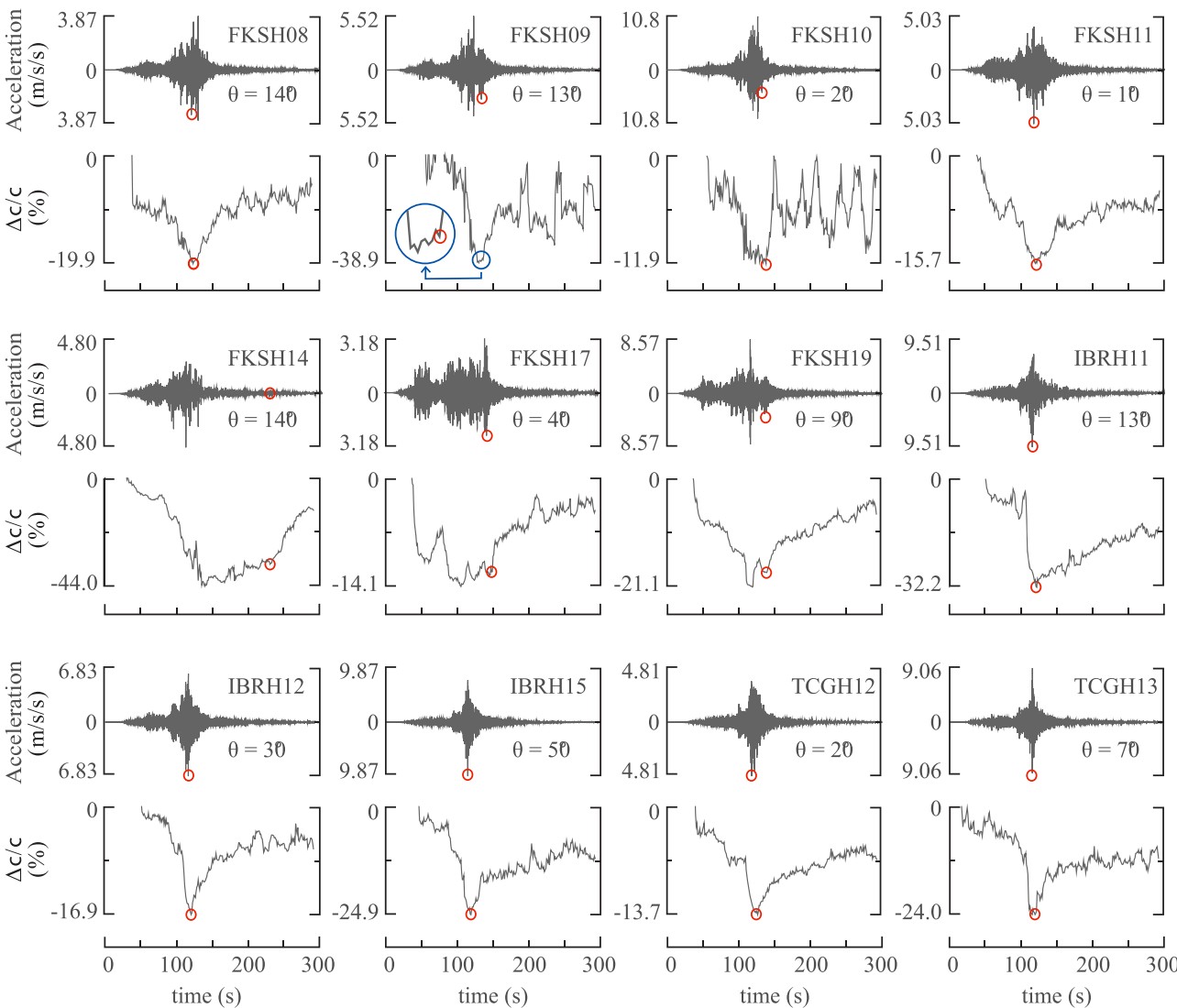

**Fig. 2 Horizontal seismograms of the Tohoku-Oki Earthquake at the selected KiK-net stations and corresponding coseismic variations of near-surface shear-wave velocity.** The horizontal seismograms are obtained by rotating the north–south (NS) and east–west (EW) components from 10° to 180° in 10° increments, in which the azimuth is measured clockwise from the north. Here shows the horizontal seismograms with the maximum PGA among the 18 directions. The maximum PGA is given by the tick label of the acceleration-axis, and the corresponding azimuth ($\theta$) is listed in each subplot. The lower tick label of the velocity-change-axis ($\Delta c/c$) represents the maximum velocity reduction (by percent) during the mainshock compared with the preseismic level of the seismic velocity before the Tohoku-Oki Earthquake. The red circle indicates the initiation of the recovery process.

the averaged normalized near-surface velocity recovery process after the Tohoku-Oki Earthquake: approximately two-thirds (62%) of the initial velocity reduction was recovered in the first stage that lasted for about 200 s, and then the velocity has increased at a much lower rate and is expected to return to the pre-seismic level after 14.7 years since the earthquake took place. These findings are quite similar to the slow recovery process of Berea sandstone in laboratory experiments by Shokouhi et al. [35] (Fig. 3c). The experimental setup mainly includes a cylindrical sample of Berea sandstone with 2.54 cm in diameter and 15 cm in length, a pump system consisting of a piezoelectric ceramic disk and a miniature accelerometer, and a probe system consisting of a pair of ultrasonic sender and receiver. In the experiments, the cylindrical sample was driven by the pump system using a 1-s-long sinusoidal excitation at the sample's fundamental resonance frequency (about 4.5 kHz) with different normal strain amplitudes, after which the change of the compressnial-wave velocity in the sample was monitored by the probe system. The normalized compressional-wave velocity recovery process in

Berea sandstone also showed two stages of logarithmic recovery, in which the first stage began from 0.01 s after the excitation signal dissipated and ended at 5.32 s, and the second stage lasted for about 200 min (slightly more than $10^4$ s).

Comparing the normalized recovery process of the near-surface shear-wave velocity from the field observations and that of the compressional-wave velocity in Berea sandstone from the laboratory experiments, we find remarkable similarities between them. First, the normalized near-surface shear-wave velocity recovery process varies slightly for different stations of which the shear strain amplitude ranges from $3.9 \times 10^{-4}$ to $3.6 \times 10^{-3}$ during the main shock (Supplementary Table 2), and the normalized compressional-wave velocity recovery process in Berea sandstone is essentially independent of the normal strain amplitude that ranges from $3.7 \times 10^{-6}$ to $7.3 \times 10^{-6}$ in the experiments[35]. Second, for both normalized recovery processes, approximately two-thirds of the initial velocity reduction is recovered quickly in the first stage, while the remaining one-third is recovered slowly in the second stage. Third, the duration of the

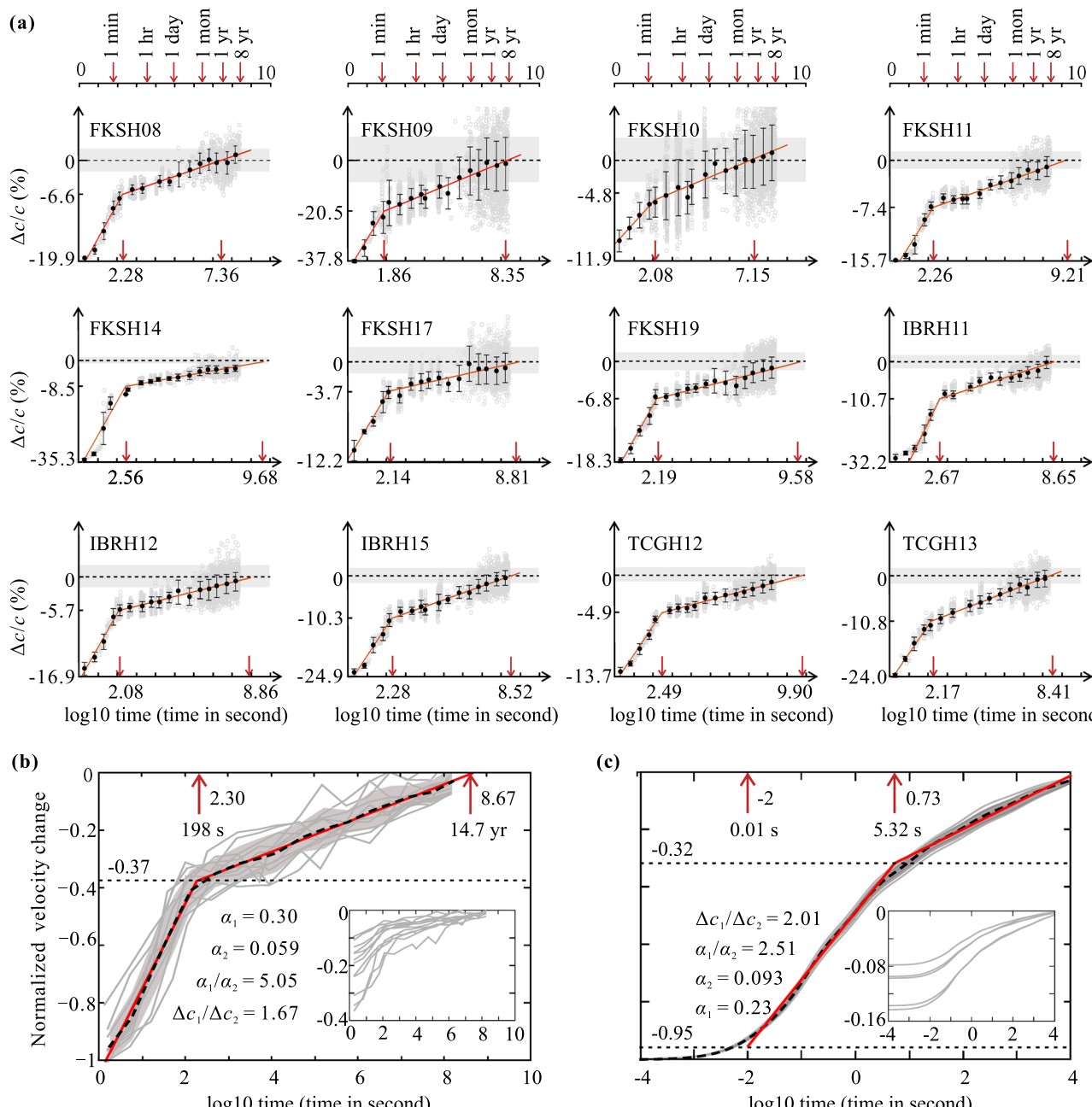

**Fig. 3 Recovery processes of seismic velocities detected in field observations and laboratory experiments. a** Recovery process of near-surface shear-wave velocity for the selected KiK-net stations. The gray dots represent raw data computed on time windows having peak acceleration lower than 0.2 m/s². The horizontal dashed line and gray-shaded band show the mean and standard deviations of the velocities before the Tohoku-Oki Earthquake. The black dots and the error bars indicate the mean and standard deviations for velocity changes ($\Delta c/c$) measured over a time span of 0.5 in the logarithmic scale. The bi-linear red lines are the regression lines of the mean values. The fitting of the first recovery stage is obtained from the first five data points for each stations except for IBRH11, for which the first two points are ignored for fitting. The fitting of the second recovery stage is obtained from the rest of data points. The details regarding the regression are provided in Supplementary Table 1. In each subplot, the red arrow on the left shows the abscissa of the intersection of the two regression lines, whereas the red arrow on the right shows the abscissa of the intersection between the regression line of the second stage and the preseismic velocity level. **b** Normalized shear-wave velocity recovery process from field observations. The red bi-linear lines and arrows are the same as in **a**. The black dotted line and gray-shaded band represent the averaged normalized velocity recovery process and the corresponding 63% confidence limit, respectively. $\alpha_1$ and $\alpha_2$ are the log slopes in the first and the second stages, respectively; $\Delta c_1$ and $\Delta c_2$ are the recovery amounts of the shear-wave velocity during the first and the second stages, respectively. The relative velocity changes for different stations before normalization are shown in the bottom right corner. **c** Normalized compressional-wave velocity recovery process from laboratory experiments (modified from Shokouhi et al.[35]). The red arrow on the left indicates the initiation of the first stage of the recovery process.

first recovery stage is generally several (about 2–6) times of the excitation time, in which the excitation time is about 1–2 min for the case of the the Tohoku-Oki Earthquake (from S wave arrival to the start time of the healing process). However, a notable difference between the two normalized recovery processes is the recovery rate (slope) in the second stage. The slope of the second stage in the field observations is much lower than that in the laboratory experiments, leading to a significantly longer second recovery stage for the case of the Tohoku-Oki Earthquake. This may be related to the scale of the target material, dynamic strain amplitude, excitation time, confining pressure and geomaterial type. More experimental data and a broader quantitative comparison between laboratory experiment and field observation will be needed to further validate the present theoretical model and establish its limitations.

**Theory**. For consolidated (bonded) granular geomaterials, seismic waves are commonly insufficient to damage the crystal structure of mineral grains, but may cause changes in inter-grain contact. Hence, the interaction between mineral grains is the basis for a physical mechanism underlying the dynamic elastic properties of consolidated granular geomaterials. In general, the inter-grain interaction mainly includes two types of contact forces, that is, elastic force due to the compression deformation of grains and adhesive force caused by surface forces of various nature[36]. The relative contribution of these two forces depends on the contact thickness[37]. For near-surface consolidated geomaterials considered here, due to a small contact thickness under relatively low confining pressure, the adhesive force plays a major role in the equilibrium state[38].

In the analysis of the adhesive force, a Lennard–Jones potential is widely adopted to describe the interaction between atoms as a function of interatomic distance, including a repulsive inverse 12-power term (Born–Mayer interaction) and an attractive inverse six-power term (van der Waals interaction), given by

$$\omega(r) = 4\eta \left[ \left( \frac{\sigma}{r} \right)^{12} - \left( \frac{\sigma}{r} \right)^{6} \right] \quad (1)$$

where $\eta$ is the binding energy, $\sigma$ is the zero-potential distance, $r$ is the interatomic distance.

Because the grain size (usually at the micrometer scale) is much larger than the inter-grain distance (usually at the nanometer scale), the atomic interactions between two grains can be represented by surface interactions. Following Yu and Polycarpou[39], the total Lennard–Jones potential energy per unit area between two half-spaces can be written as

$$\psi_1(z) = -\frac{4\pi^2 \rho_1 \rho_2 \eta \sigma^6}{12\pi z^2} \left[ 1 - \frac{1}{30} \left( \frac{\sigma}{z} \right)^6 \right] \quad (2)$$

where $z$ is the distance between the surfaces; $\rho_1$ and $\rho_2$ are the number densities of the atoms of the two interacting bodies. Using the definition of the Hamaker constant $A = 4\pi \rho_1 \rho_2 \eta \sigma^6$, Eq. (2) could be rewritten as

$$\psi_1(z) = -\frac{A}{12\pi z^2} \left[ 1 - \left( \frac{z_0}{z} \right)^6 \right] \quad (3)$$

where $z_0 = (1/30)^{1/6}\sigma$ is the equilibrium distance between two surfaces.

According to the existing adhesive contact theory (a more detailed description can be found in Lebedev and Ostrovsky[38] and Sens-Schönfelde et al.[40]), if the inter-grain distance exceeds a certain threshold under the action of external forces, the adhesive contact will be broken, resulting in a significant decrease in material modulus; and then the broken contact can be recovered at a closer inter-grain distance during unloading, leading to

stress-strain hysteresis. Moreover, as demonstrated by Lebedev and Ostrovsky[38], the adhesive force potential should have at least one additional local minimum (defining the metastable state) besides the global minimum (defining the stable state) to account for the slow dynamics of bonded granular geomaterials. They pointed out that a fraction of the broken contacts can remain trapped in the metastable state after the break, and will overcome the potential barrier by thermal motion to return to the stable state over a long relaxation period, which generates slow dynamics.

Referring to the DLVO theory used in the explanation of the stability of colloidal suspension, Lebedev and Ostrovsky[38] employed a superposition of the Lennard–Jones potential energy and the potential energy of the electrical double-layer interaction between surfaces to construct a double-well potential energy model. However, we found that in their potential energy model the second metastable minimum requires a fairly high ion concentration and is quite far from the initial equilibrium position, indicating that the DLVO theory may not be applicable to the slow dynamics of consolidated granular geomaterials. This is because that the electrical double-layer interaction between surfaces is a weak, long-range force that decays exponentially with distance with a characteristic decay length equal to the Debye length. The Debye length decreases as the ion concentration increases, and is 960 nm in pure water, typically 1–100 nm in aqueous solutions. For example, for monovalent solution, the Debye lengths are about 100, 10, and 1 nm at $10^{-5}$, $10^{-3}$, and $10^{-1}$ mol/L, respectively.

In this study, considering the widespread existence of water in near-surface geomaterials, we proposed a multiple-well potential energy model that includes a concurrence between the Lennard–Jones potential energy and the potential energy of the hydration interaction between surfaces[37], given by

$$\psi(z) = -\frac{A}{12\pi z^2} \left[ 1 - \left( \frac{z_0}{z} \right)^6 \right] - \frac{kT}{d^2} \cos\left( \frac{2\pi z}{d} \right) \exp\left( -\frac{z}{d} \right) \quad (4)$$

where $k = 1.38 \times 10^{-23}$ J/K is the Boltzmann constant; $T$ is the Kelvin temperature, set as 296 K in this study to keep in line with the temperature controlled in the experiments by Shokouhi et al.[35]; and $d$ is the diameter of a water molecule. As shown in the second term in Eq. (4), the hydration interaction between two solid surfaces is a decaying oscillatory (sinusoidal), short-range force, which has been measured directly using x-ray reflectivity techniques since about two decades ago[41], and is considered to originate from the fluctuations of the liquid densities between two solid surfaces at nanometer separations[37].

Here we consider an example of the resulting potential energy for two quartz surfaces with sandwiched water (Fig. 4), in which the Hamaker constant $A = 1.36 \times 10^{-20}$ J is calculated based on the Lifshitz theory[37]; the equilibrium distance $z_0 = 0.895$ Å is estimated by the sizes of atoms[39]; and the diameter of a water molecule $d = 3.85$ Å is obtained from the volume of a water molecule.

$$A = \frac{3}{4}kT \left( \frac{\eta_1 - \eta_2}{\eta_1 + \eta_2} \right)^2 + \frac{(2\nu_1 + \nu_2)h}{16\sqrt{2}} \frac{(n_1^2 - n_2^2)^2}{(n_1^2 + n_2^2)^{3/2}} \quad (5)$$

$$z_0 = \frac{1}{2} \times \left( \frac{1}{60} \right)^{1/6} \times (\sigma_1 + \sigma_2) \quad (6)$$

$$d = \left( \frac{6m}{\pi \rho_w N_A} \right)^{1/3} \quad (7)$$

where $\eta_1 = 4.29$ and $\eta_2 = 79.02$ are the dielectric permittivities of quartz and water, respectively; $\nu_1 = 3.234 \times 10^{-15}$ s$^{-1}$ and $\nu_2 =$

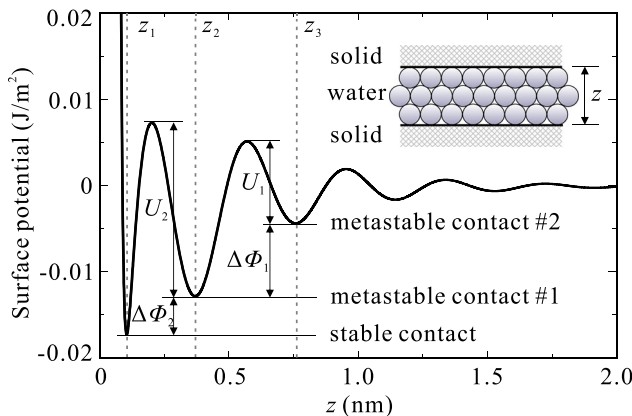

**Fig. 4 Contact force potential for two quartz surfaces with sandwiched water.** $z_1$, $z_2$, and $z_3$ are the contact distances corresponding to the stable state, the first and the second metastable states, respectively; $U_1$ and $U_2$ are the heights of the potential barriers in the first and the second stages, respectively; $\Delta\psi_1$ and $\Delta\psi_2$ are the variations in the surface potential energy of the first and the second stages, respectively.

$3.00 \times 10^{-15}$ s$^{-1}$ are the main electronic absorption frequencies in the ultraviolet of quartz and water, respectively; $n_1 = 1.536$ and $n_2 = 1.333$ are the refractive indexes of quartz and water in the visible, respectively; $\sigma_1 = 1.11$ Å and $\sigma_2 = 0.66$ Å are the covalent radii of silicon and oxygen atoms, respectively; $m = 18$ g is the mass of 1 mol water, $N_A = 6.02 \times 10^{23}$ is the Avogadro constant, and $\rho_w = 1.0 \times 10^3$ kg/m$^3$ is the density of water.

Slow relaxation due to stochastic dynamics occurs in a diversity of physical processes, such as the aging of various glasses[42], the evolution of frictional strength[43], and the relaxation of crumpled surfaces[44]. In general, for a system governed by thermal activation, the process for the system to cross the energetic barrier to reduce its energy is considered to be a Markov process, and the reaction rate is given by the Arrhenius equation in chemistry; namely, $\lambda \propto e^{-U/kT}$, where $\lambda$ and $U$ are the reaction rate and the height of the energetic barrier, respectively. In this study, for consolidated granular geomaterials, following the work in Lebedev and Ostrovsky[38] and Ostrovsky et al. [45], the recovery of the elastic modulus during the metastable contacts return to the stable state can be described by

$$\frac{dE_s}{dt} = -BE_0 \exp(-E_s/\Lambda) \tag{8}$$

where $E_S$ is the decrease in modulus due to the formation of the metastable contacts; subscript "S" represents slow dynamics; $t$ denotes the time; $E_0$ is the unperturbed elastic modulus; $B$ is a constant-factor determining the rate of recovery; $\Lambda = kT/V$, and $V = U/(\delta E_0)$ is the characteristic volume of the metastable contact, where $\delta$ is the initial fraction of the metastable contacts.

Solving Eq. (8) with the boundary conditions, namely, $E_S(0) = E_A$ and $E_S(\tau) = 0$, we obtain

$$E_S = \Lambda \ln\left(-\frac{BE_0}{\Lambda}t + \exp(E_A/\Lambda)\right) \tag{9}$$

in which, $B = -\Lambda/[(E_0\tau)(1 - \exp(-|E_A/\Lambda|))]$, and $\tau$ is the time for all the metastable contacts to return to the stable state. According to $E = \rho c^2$ ($\rho$ denotes the density), we can get

$$\Delta c/c = E_S/2E_0 = \frac{\Lambda}{2E_0}\ln\left(\frac{t}{\tau} + \left(1 - \frac{t}{\tau}\right)\exp(-|E_A/\Lambda|)\right) \tag{10}$$

This equation implies that in the beginning of the recovery, for small $t << \tau \exp(-|E_A/\Lambda|)$, we have

$$\Delta c/c = -|E_A|/2E_0 = \text{constant} \tag{11}$$

and then, due to $E_A/\Lambda << 1$, there exists a long-term logarithmic

recovery:

$$\Delta c/c = \frac{\Lambda}{2E_0}\ln\left(\frac{t}{\tau}\right) \tag{12}$$

Here we discuss a two-stage recovery process based on the multiple-well potential energy model proposed. Clearly, the three deepest potential wells with the closest inter-grain distances define the stable contact, and the first and the second metastable contacts, respectively. In this case, considering that the duration of the first stage of the recovery process is much shorter than that of the second stage, the first stage corresponds to overcoming the potential barrier between the first and the second metastable contacts, and the second stage corresponds to overcoming the potential barrier between the first metastable contact and the stable contact. Therefore, regarding the first metastable contact as a temporary stable contact in the first stage, both stages in the recovery process follow an approximate logarithmic recovery. Moreover, the ratio of the log slope in the first stage to that in the second stage can be estimated according to Eq. (12):

$$\frac{\alpha_1}{\alpha_2} = \frac{U_2}{U_1} = 2.13 \tag{13}$$

in which, $\alpha_1$ and $\alpha_2$ are the log slopes in the first and the second stages, respectively; $U_1 = 9.50 \times 10^{-3}$ J/m$^2$ and $U_2 = 2.02 \times 10^{-2}$ J/m$^2$ are the heights of the potential barriers in the first and the second stages, respectively.

As described by Ostrovsky et al. [45], the decrease in modulus can also be expressed by $E_A = P_S/\varepsilon$, in which $P_S$ is the equivalent stress due to the deformation of the metastable contact, and $\varepsilon$ is the equivalent strain of the metastable contact. According to the first law of thermodynamics, the variation of the surface potential energy equals to the work done by the equivalent stress, given by

$$\Delta\psi = \int_0^{z_m} P_S \, dz = \frac{1}{2}E_A \varepsilon_m z_m \tag{14}$$

in which, $z_m$ is the characteristic thicknesses of the metastable contact, defined as the difference in contact distances between the metastable and stable states; $\varepsilon_m$ is the corresponding strain of the metastable contact to $z_m$. From Eq. (14), we have

$$\frac{\Delta c_1}{\Delta c_2} = \frac{E_{A1}}{E_{A2}} = \frac{\Delta\psi_1 z_{m1}^2 z_3}{\Delta\psi_2 z_{m2}^2 z_2} = 0.96\frac{\Delta\psi_1}{\Delta\psi_2} = 1.88 \tag{15}$$

in which, $\Delta c_1$ and $\Delta c_2$ are the recovery amounts of the seismic velocity in the first and the second stages, respectively; $E_{A1}$ and $E_{A2}$ are the recovery amounts of the shear modulus in the first and the second stages, respectively; $\Delta\psi_1$ and $\Delta\psi_2$ are the variations in the surface potential energy of the first and the second stages, respectively; $z_1 = 1.05$ Å, $z_2 = 3.71$ Å, and $z_3 = 7.59$ Å are the contact distances corresponding to the stable state, the first and the second metastable states, respectively; $z_{m1} = z_2 - z_1$ and $z_{m2} = z_3 - z_2$ are the characteristic thicknesses of the first and the second metastable contacts, respectively. In such case, the threshold strain for nonequilibrium dynamic behavior can be estimated as $\varepsilon_S \approx z_m/R_g \approx 10^{-7}$ ($R_g \approx 0.2$ mm is a typical grain size for sandstone), which is in good agreement with the experimental results by Pasqualini et al. [8]. Additionally, both Eqs. (13) and (15) are well consistent with the experimental results by Shokouhi et al. [35] (Fig. 3c).

## Discussion

Elastic wave velocity is an indicator of modulus, hence our observation demonstrates that a great earthquake can cause significant softening and long-term healing in the near surface over a wide area (Figs. 2 and 3a). For those 12 selected stations, the maximum coseismic velocity reduction in the upper 100–200 m during the 2011 Mw9 Tohoku-Oki earthquake ranged from 11.9% to 46.3%.

Note that the velocity reduction is unevenly distributed in vertical direction, and it is probably that most of the velocity reduction is localized in shallow layers[27,46]. This is because the shallow layers have much lower velocities than the deep layers (Supplementary Fig. 3), and the elastic nonlinearity of granular materials decreases with increasing confining pressure[7]. Hence, the shallow soft sediments of the top tens of meters play a major role in the near-surface softening, and a greater velocity reduction can be expected in the shallow subsurface compared to the average velocity reduction from the ground surface to the borehole bottom. Then we find that about two-thirds of the initial reduction is recovered rapidly in a few minutes, and the remaining one-third is recovered slowly over months or years (Fig. 3b, c). It also should be noted that the long-term recovery process has not been taken into account in traditional seismic hazard yet. These findings indicate that in site-specific seismic hazard estimates, it may be necessary to reevaluate the seismic site response after a large earthquake occurs to consider the effects of long-term changes in near-surface stiffness caused by strong perturbations. Finally, based on the work of Ostrovsky et al.[45], we propose a theoretical multiple-well model of the contact force potential between grains in geomaterials, and describe the slow relaxation as the transition process from a metastable contact to another metastable contact or the stable contact (Fig. 4).

However, many details of the physical mechanism remain unclear. For example, the first recovery stage in the field observations is not entirely a strictly slow-dynamics process, because the seismic wave has not yet ended when the first recovery begins. In this case, the fast dynamics and slow dynamics can occur simultaneously (Supplementary Fig. 4). In a recent study, Ostrovsky et al.[9] developed a modified version of the Arrhenius theory to simultaneously describe nonlinear fast and slow dynamics under sinusoidal dynamic wave excitation. Nevertheless, the amplitude of the sinusoidal dynamic wave excitation considered in their theory and used in the experiment does not vary with time. It is still an open question how the velocity varies under variable amplitude dynamic wave excitation in both theory and lab experiment. We believe that the softening and healing of geomaterials should not be limited to the scope of seismology and geophysics but involves multiple disciplines, and further studies are needed in the future.

## Methods

**Station and record selection**. The Kiban-Kyoshin strong-motion observation network (KiK-net) in Japan currently consists of 698 seismic stations, which was first established by the National Research Institute for Earth Science and Disaster Resilience (NIED) after the 1995 Mw6.8 Hanshin-Awaji Earthquake[47]. These stations are uniformly distributed throughout Japan with an inter-station distance of about 25 km. Each KiK-net station has a borehole of at least 100 m in depth, and a pair of three-component seismographs have been installed on the ground surface and at the bottom of the borehole. The sampling frequency of the seismographs was originally set to 200 Hz, and has been changed to 100 Hz since 2007. The frequency response of the seismographs is flat from DC to about 20 Hz[48]. The total length of the seismograms is 60–300 s which includes 15 s pre-trigger data. For most of the boreholes, the geological information and velocity profiles are also available on the NIED website.

In the 2011 Mw9.0 Tohoku-Oki Earthquake, a total of 59 KiK-net stations captured strong ground motions with peak accelerations exceeding 2 m/s², in which a peak acceleration of 1–2 m/s² has commonly been cited as a threshold that causes obvious nonlinear site effects[31,49]. Out of these 59 stations, 12 stations were selected to investigate the near-surface softening and healing associated with the 2011 Mw9.0 Tohoku-Oki Earthquake according to the following selection criteria (Supplementary Fig. 5, details about the selected stations are provided in Supplementary Table 3). First, since our target is the near-surface region, the borehole depth for the selected stations should be no more than 250 m (52 stations fulfill this criterion). Then, we excluded 22 stations whose near-surface seismic velocities were significantly affected by other earthquakes in the study period, 16 stations whose near-surface seismic velocities suddenly changed due to instrumental replacement or relocation (Supplementary Fig. 6), and 2 stations whose near-surface healing processes were masked by strong seasonal variations (Supplementary Fig. 7).

Out of the 12 selected stations, 10 stations have been operating since 2000, and 2 stations (i.e., IBRH12 and IBRH15) have been operating since 2003. As of the mid of March 2019, the 12 stations had collected 27,774 sets of seismograms in

total (each set includes six components, that is, three components at both surface and borehole). We divided the time span into three periods: from the operation start time to 14:46:24 11 March 2011 (the reference period, before the Tohoku-Oki Earthquake), from 14:46:24 11 March 2011 to 18:33:04 12 March 2011 (period 1, within 10⁵ s of the Tohoku-Oki Earthquake), from 18:33:04 12 March 2011 to 18 March 2019 (period 2, from 10⁵ s to 8 years after the Tohoku-Oki Earthquake). With the exception of the seismograms recorded in period 1, we limited the value of surface horizontal peak acceleration to smaller than 0.2 m/s², in which 0.2 m/s² corresponds to a conservative estimate of the threshold of the emergence of slight nonlinear site effects[30,49]. In addition, to ensure that the seismic waves propagate in a near-vertical direction between the borehole and surface seismograph, we selected seismograms with focal depths of at least 7 km[11]. Finally, a total of 23,306 sets of seismograms from 3762 earthquakes were used in this study (Supplementary Fig. 8), and the numbers of the seismograms in each period are 7308, 163, and 18,310, for the reference period, period 1, and period 2, respectively.

**Seismic interferometry**. Deconvolution-based seismic interferometry has proven to be a powerful tool in monitoring near-surface seismic velocities with high precision for vertical borehole arrays, which uses the deconvolved waveforms of seismograms recorded at different depths to extract the seismic wave propagation time between them[11,50,51]. The deconvolution function in the frequency domain is given by

$$D(x_s, x_b, \omega) = \frac{u(x_s, x_r, \omega)}{u(x_b, x_r, \omega)} \approx \frac{u(x_s, x_r, \omega) u^*(x_b, x_r, \omega)}{|u(x_b, x_r, \omega)|^2 + \chi} \quad (A1)$$

where $\omega$ is the frequency, * represents the complex conjugate; $u(x_b, x_r, \omega)$ is the Fourier spectra of the incoming wavefields recorded at the bottom of the borehole located at $x_b$, from a source located at $x_r$, and $u(x_s, x_r, \omega)$ is the Fourier spectra of the corresponding wavefields recorded at the ground surface located at $x_s$; $|u(x_b, x_r, \omega)|^2$ is the power spectra of the incoming wavefields smoothed by the multitaper method with a time–half-bandwidth product of 4[27,52]; $\chi$ is a regularization parameter to ensure non-zero values in the denominator, set as 1% of the average power spectrum of the incoming wavefields in the frequency range of 1–13 Hz.

We first removed the pre-event noise from the seismograms after picking the P wave arrival, and then rotated the north–south (NS) and east–west (EW) components of the seismograms from 10° to 180° in 10° increments to synthesize horizontal seismograms in 18 directions, in which the azimuth is measured clockwise from the north. For seismograms recorded within 10⁵ s of the Tohoku-Oki Earthquake, we used short-time moving windows that are 10.24 s long each and overlap by 90%, while the time window used for other seismograms starts from the onset of the P wave until the end of the signal. After applying a 2.5% Hanning taper to each time window, we calculated the deconvolution function in the frequency domain according to Eq. (A1) and took the inverse Fourier transform in the frequency range of 1–13 Hz to obtain the deconvolved waveforms in the time domain. Next, we resampled the deconvolved waveforms at intervals of 0.001 s using a cubic spline interpolation to enhance their time resolution. The wave propagation time can be obtained as the peak time of the interpolated deconvolved waveforms near the reference traveltime based on the velocity profile. Thereafter, the near-surface seismic velocity was expressed as a function of azimuth angle by dividing the borehole depth by the wave propagation time in different directions. Following Nakata and Snieder[51], the seismic velocity can be divided into isotropic and anisotropic terms using Fourier series expansion, in which the isotropic term is calculated as the average velocity in the 18 directions, and the anisotropic term is resulted from shear wave splitting. In this study, the splitting time was much smaller than the traveltime for all the selected stations; therefore, the isotropic term was chosen to represent the near-surface velocity.

## Data availability

Seismic data used in the study are provided by National Research Institute for Earth Science and Disaster Resilience[47] and are available at: http://www.kyoshin.bosai.go.jp/. Source data are available on Zenodo: https://zenodo.org/record/4326234[53].

## Code availability

The Matlab codes for reproducing the main results of this study are available on Zenodo[53]: https://zenodo.org/record/4326234

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

## Acknowledgements

We thank Fei Han at Purdue University for his fruitful suggestions on our manuscript. This study was financially supported by National Science Foundation of China (Nos. 52008184, 51978304, 51908237, and 51778260), and China Postdoctoral Science Foundation (2018M642845). We are grateful to the National Research Institute for Earth Science and Disaster Resilience (Japan) for providing us with the KiK-Net data.

## Author contributions

Y.M., S.-Y.W., and H.-Y.Z. conceived and supervised this project. S.-Y.W. and H.Z. carried out seismic interferometry calculations. S.-Y.W. and H.-J.H. carried out theoretical analysis. S.-Y.W. wrote the manuscript. W.-P.J, E.-L.Y., B.R., and Y.-X.W. discussed the results and assisted during manuscript preparation.

## Competing interests
The authors declare no competing interests.
