## [Peer Review File · Nature Communications]

REVIEWER COMMENTS

General comments

This paper consists of mainly two parts: one is the field study of near surface velocity change associated with the 2011 Tohoku-Oki Earthquake, and another is the theoretical study to account for the observed two-stage recovery process of seismic velocity. On the field study, I basically agree with the main conclusion although I have some concerns in the methodology and interpretation of the results. On the other hand, although I do not think I understand the detail, the logic of the theoretical part seems like a leap forward. I think the first recovery stage (~ 200 s) observed in the field study merely reflects the classic nonlinearity of shear modulus (dependence of shear modulus on amplitude of dynamic strain). I think the transition from the metastable contact #1 to the stable contact could be a possible mechanism for the second recovery stage, however, it is questionable if the proposed model (transition from the metastable contact #2 to #1) can be applied to the first recovery stage. Nevertheless, the authors addressed a scientifically important issue and found novel and reliable results that may contribute to this area. As we do not know much about the model of the recovery process, I think just presenting a theory is worth it, regardless whether it matches the observation. In individual comments, I list up some concerns about the methodology and interpretation of the results, and request some minor corrections to improve quality of the draft.

Individual comments

1. Fig.1: On the map inserted in upper left corner, there is no particular reason why four Japanese islands are colored.
2. Fig.2: Show the component (EW or NS) of the seismogram. Write a horizontal dashed line corresponding to the preseismic level. The time scale (abscissa) should be shown for all subfigures.
3. P6 "In order to \sim computational efficiency": I wonder why the moving-window analysis is applied to all the seismograms recorded in period 1. I understand it is necessary to improve temporal resolution of the recovery process for the mainshock seismogram (Fig. 2). However, this procedure may not be necessary (non-moving time window is enough) for other earthquakes occurring within 28 hours. The similar thing can be said in P12 "With the exception of \sim smaller than 0.2 m/s² ": I don't know why the threshold of 0.2 m/s² is not applied to the seismograms obtained in period 1. This threshold seems to be important for all the periods.
4. P6 "As of March 2019 \sim permanent velocity change": I think this is an overstatement as no stations show an excess recovery over the preseismic level beyond the confidence interval at 8 yr in Fig. 3a.
5. P6 "We also noticed that \sim largest relative velocity changes": Are there any evidences of cyclic mobility in waveforms of IBRH11?
6. The time scale of the first recovery stage (about 200 s) is comparable to the duration of the strong ground motion of the Tohoku-Oki Earthquake (Fig. 2a). I think this indicates that the first recovery merely reflects the classical nonlinear elasticity: as the amplitude of coda wave attenuates, the shear modulus recovers. To confirm this hypothesis, I recommend the authors to make a plot of the ground motion level versus the velocity reduction ratio for the mainshock seismogram, which shows the amplitude-dependence of the velocity (classical nonlinear elasticity). If the plotted curve behaves like a multivalued function (more than two values for a given ground motion level), the difference between the two velocity reduction values may be attributed to the non-classical nonlinearity.
7. P7 "This may be \sim requires further study": I think scale of the target material is an important factor that determines duration of the second recovery. The scale of soil layer (~ 102 m) is much

larger than the sample of Berea sandstone (~10-1m), which may result in longer recovery time for the former.

8. Fig.3b,c: For the field observation (Fig. 3b), the two stage log-linear recovery is clearly observed. However, for the Shokouhi et al. (2017)'s experiment (Fig. 3c), the recovery curve does not show a clear bi-linear function if the red regression line is removed (see the inserted panel).

9. P8 "According to the existing ~ stress-strain hysteresis": If my understanding is correct, Sens-Schonfelder et al (2019, <https://academic.oup.com/gji/article/216/1/319/5116168>) proposed a model to explain the nonlinear elasticity based on the similar concept.

10. The authors argue that the first recovery stage corresponds to the potential jump from the metastable contact #2 to #1. I think this idea is not appropriate to explain the results from the KiK-net study because of the reasons commented in #6. For the Shokouhi et al. (2017)'s experiment and the second recovery of the KiK-net results, the proposed idea (the potential jump from the metastable contact #1 to the stable contact) sounds reasonable.

11. Conclusions: Conclusion should focus on what authors did in this study, not the study of other researchers. The sentence of "For example ~ Vucetic and Dobry (1991)" is not suitable for the conclusion.

12. Station and record selection: I think the finally selected 12 stations are also influenced by strong earthquakes. Seismicity of the target region activated greatly after the Tohoku-Oki Earthquake, which includes Mj7.0 event occurred on April 11, 2011 (see <https://agupubs.onlinelibrary.wiley.com/doi/full/10.1029/2012GL051491>). Therefore, I'm concerned about contamination of another velocity reduction and recovery due to these large earthquakes.

13. Station and record selection: In the selection of analyzed seismograms, S/N ratio should be much larger than one within the target frequency range because noise is mostly composed of surface wave and assumption of vertical incidence (must be satisfied in the deconvolution interferometry) is not appropriate for noise.

Reviewer #2 (Remarks to the Author):

The authors investigated the near-surface material softening process after the 2011 Tohoku earthquake using seismic interferometry. The seismic interferometry uses co-located seismic stations at different depths from the KiK-net. The differential travel time for a given earthquake between the two stations at the same site is used to estimate the local shear-wave velocity at the near-surface. This is a standard technique and has been proven useful to monitor near-surface velocity changes after large earthquakes. The authors worked on 3762 earthquakes to obtain an eight-year time series of the velocity reduction-recovery process in the Honshu region after the Tohoku earthquake. They found a clear velocity reduction at all sites due to the Tohoku earthquake that occurred during the passage of the seismic waves and a two-stage recovery process that is still ongoing to date. The first stage healed about 60% of the velocity reduction within hundreds of seconds and the second stage may take up to 15 years to fully recover to the previous local strength. To understand the process, the authors compared the observations to a laboratory experiment and analytical models. The laboratory experiment that the author cited also demonstrates a two-stage recovery process that is similar to the field observations. With the assumed model setup, which includes two solid surfaces sandwiching a water layer, the authors obtained solutions that are consistent with the field observations.

It is a very interesting paper and I enjoyed reading it. I am not an expert of the inter-grain interaction modeling, so I will defer it to other reviewers' expert opinions. The seismic data processing is rigorous and well-reasoned. In particular, I appreciate the author systematically examined the shear-wave velocity recovery process over eight years. However, the comparison

between the laboratory experiment and the field observations requires more justifications. The laboratory experiment investigated the P-wave velocity evolution process, and it is unlikely to be the same as that of S-waves. Besides, the shear strains of the field observations are two to three orders larger than the laboratory environment. Assuming the recovering process depends on the state and the rate as the analytical modeling suggests, this large deviation may lead to different processes. It is always challenging to scale lab results to field observations, and the apparent agreement between the experiment and the observations may not be consistent.

Reviewer #3 (Remarks to the Author):

I have reviewed the paper entitled "Near-surface softening and healing in eastern Honshu associated with the 2011 magnitude-9 Tohoku-Oki Earthquake" by Su-Yang Wang, Hai-Yang Zhuang, Hao Zhang, Hong-Jun He, Wei-Ping Jiang, Er-Lei Yao, Bin Ruan, Yong-Xin Wu and Yu Miao. The paper deals with interesting co-seismic phenomena that involve material softening and subsequent healing. This behavior has already been observed for a long time in rock and concrete, but it is relatively new in soils. The authors propose some theoretical hypotheses to explain such observations in records from the 2011 M9 Tohoku earthquake. The paper is in general well written and clear. However, I have some comments that I hope will help to improve this interesting work. In any case, I support the publication of this article after the corrections are made.

General comments

This paper uses records from the 2011 M9 Tohoku earthquake to study the nonlinear behavior of shallow material, namely the softening and healing processes during strong shaking. The authors use seismic interferometry to estimate the velocity changes from surface and borehole records. This technique is well described in Nakata and Snieder (2012) and Bonilla et al. (2019), yet this technique is mentioned without telling what it measures (i.e. travel time), and since it is the base for the discussion, I think it deserved more explanation in the text body or say that the details will be seen in the supplementary material. For example, time window length, the particular formulation of the seismic interferometry (you use multitaper deconvolution), overlapping between windows. And a very important information that is missing is how the velocity change is computed? What components do you use (NS, EW, UD, a combination of them)? What is the frequency band that you study at each station, and how do you estimate it? As it is written, it seems that this operation is trivial, and it is not and this has been shown in the references above. In other words, you need to convince the readers that your computations are stable before analyzing their results.

Another issue is the earthquakes used in this study. The time span is at least 8-9 years after the 2011 mainshock. Are the events coming from the same epicentral area or they come from all over Japan? Is there any depth constraint? Do you use the same events for all stations? or you have a set of earthquakes for each station. Once this is done, how do you estimate the error bars in the velocity changes? Qin et al. (2020) compute the error bars from the statistics of the travel times when using either seismic interferometry or autocorrelation functions.

There is no mention on how the mean velocity (and the error estimate) is computed at each station. Given the numbers, I suppose it is the average shear wave speed for the whole soil column. The problem that you might have with geotechnical people is that the numbers are relatively large, and a strong velocity change is reported. Thus, in their point of view, how a material that is close to 900 m/s can change up to 20%. Since I worked at station IBRH16 (Bonilla et al., 2019), and due to the use of autocorrelation functions and the knowledge of the velocity profile, it is probably the first tens of meters that produced such velocity reduction because deeper layers have larger shear velocities. In other words, the nonlinear behavior is rather shallow and localized in the softer soils close to the surface. I recommend to write something about this because the theory that is used is for granular material, which is more likely to happen in the

shallow part only. I would probably plot the velocity profiles to discuss this issue.

Regarding the hypothesis that pore pressure and in particular, dilatancy effects might happen, I am not sure how you can demonstrate this. I also recommend reading our paper Qin et al. (2020) where we studied a borehole array with pore pressure transducers, and it was difficult to relate pore pressure effects and velocity changes. In my opinion, the grain arrangement during the shaking is more important than pore pressure effects if the material is not fully saturated. However, Tohoku produced such large PGA's that both effects could be present at the same time. This is not the case with our study in California. In any case, this needs more justification, or just soften the text because it is a not easy subject. The method that you describe Kostadinov and Yamazaki (2001) basically computes the instantaneous frequency, which is also shown in Bonilla et al. (2019), but no evidence of liquefaction at the site was found. Thus, again, this is a tricky question, and the change of frequency can also be related to the change of the velocity change itself, thus it is a circular thinking.

Regarding the theory you use to explain the observations, I suggest to give a proper credit to the latest work of Ostrovsky et al. (2019). Indeed, you basically use their technique on the results of your computations. I may be missing something, but the new thing here is the application of such technique in Tohoku data, and it works quite well. Yet again, I am worry about the geotechnical community because you report velocity changes for the whole soil column and the processes can be very different at each soil condition present in the column. What is interesting for me is that there is a trending behavior, and probably all materials have it, what we see is the average, and you could write like that to avoid critics on a particular nonlinear soil behavior.

Another interesting consequence is the recovery time that takes almost 10 years for these stations. This is something that is not taken into account in traditional seismic hazard. This means, that even if we characterize well a site, the known site response is not going to reduce the ground motion variability. If a strong perturbation takes place, the system will take time to recover, and this is proportional to the amplitude of the perturbation. This is critical for site-specific seismic hazard estimates.

I hope this helps,

Sincerely,

Fabian Bonilla

Replies to Reviewer #1

General comments:

This paper consists of mainly two parts: one is the field study of near surface velocity change associated with the 2011 Tohoku-Oki Earthquake, and another is the theoretical study to account for the observed two-stage recovery process of seismic velocity. On the field study, I basically agree with the main conclusion although I have some concerns in the methodology and interpretation of the results. On the other hand, although I do not think I understand the detail, the logic of the theoretical part seems like a leap forward. I think the first recovery stage (~200 s) observed in the field study merely reflects the classic nonlinearity of shear modulus (dependence of shear modulus on amplitude of dynamic strain). I think the transition from the metastable contact #1 to the stable contact could be a possible mechanism for the second recovery stage, however, it is questionable if the proposed model (transition from the metastable contact #2 to #1) can be applied to the first recovery stage.

Nevertheless, the authors addressed a scientifically important issue and found novel and reliable results that may contribute to this area. As we do not know much about the model of the recovery process, I think just presenting a theory is worth it, regardless whether it matches the observation. In individual comments, I list up some concerns about the methodology and interpretation of the results, and request some minor corrections to improve quality of the draft.

Response: We really appreciate your detailed comments and suggestions. We have tried our best to revise the manuscript according to your comments. In the annotated version of the revised manuscript, all changes are highlighted in red. Thank you again for your careful review of our manuscript. Itemized response to your questions is appended below.

Individual comments:

1. Fig.1: On the map inserted in upper left corner, there is no particular reason why four Japanese islands are colored.

Response: Thank you for your comments. We have tried to color the four islands into gray as shown in Fig. R1. However, we found that it is difficult to mark the names of those islands on the map. Hence, the main reason why the islands are colored is to distinguish different islands and to mark the names of these four islands.

Fig. R1 Topographic map of northern Honshu Island and surrounding area, with slip distribution of the Tohoku-Oki Earthquake.

2. Fig.2: Show the component (EW or NS) of the seismogram. Write a horizontal dashed line corresponding to the preseismic level. The time scale (abscissa) should be shown for all subfigures.

Response: Thank you for your comments. We have revised Figure 2 accordingly. In our manuscript, we rotated the north-south (NS) and east-west (EW) components of the seismograms from 10° to 180° in 10° increments to synthesize horizontal seismograms in eighteen directions, in which the azimuth is measured clockwise from the north. Figure 2 shows the horizontal seismograms with the maximum PGA in the eighteen directions, and the corresponding azimuth (θ) is given in each subplot.

Figure 2: Horizontal seismograms of the Tohoku-Oki Earthquake at the selected KiK-net stations and corresponding coseismic variations of near-surface shear-wave velocity. The horizontal seismograms are obtained by rotating the north-south (NS) and east-west (EW) components from 10° to 180° in 10° increments, in which the azimuth is measured clockwise from the north. Here shows the horizontal seismograms with the maximum PGA among the eighteen directions. The maximum PGA is given by the tick label of the acceleration-axis, and the corresponding azimuth (θ) is listed in each subplot. The upper and lower tick label of the velocity-axis represent the preseismic level of the seismic velocity before the Tohoku-Oki Earthquake and the lowest velocity during the mainshock, respectively. The maximum velocity reduction (by percent) is listed in the bottom-right corner of each subplot. The red circle indicates the initiation of the recovery process.

3. P6 “In order to ~ computational efficiency”: I wonder why the moving-window analysis is applied to all the seismograms recorded in period I. I understand it is necessary to improve temporal resolution of the recovery process for the mainshock seismogram (Fig. 2). However, this procedure may not be

necessary (non-moving time window is enough) for other earthquakes occurring within 28 hours. The similar thing can be said in P12 “With the exception of ~ smaller than 0.2 m/s^2 ”: I don’t know why the threshold of 0.2 m/s^2 is not applied to the seismograms obtained in period 1. This threshold seems to be important for all the periods.

Response: Thank you for your comments. As shown in Fig. 3, the first stage of the recovery process lasts for 72-468 ($10^{1.86}$ - $10^{2.67}$) seconds at those selected 12 stations, which means that the first stage has not ended at the end of the mainshock for some stations. Therefore, it is not enough to apply moving window only to the mainshock. Having known the duration of the first stage, we acknowledged that moving-window analysis may not be necessary for other earthquakes occurring within 28 hours (one hour should be enough). Nevertheless, considering that there are only 163 seismograms recorded in period 1, it is still acceptable to apply moving-window analysis to all the seismograms recorded in period 1 (the moving-window analysis for all the 163 seismograms can be completed within one or two hours on a personal computer).

Out of these 163 seismograms, there are 47 seismograms recorded within one hour after the mainshock, and all the 47 seismograms have PGAs greater than 0.2 m/s^2 . Hence, it is not feasible to remove all the seismograms with PGA great than 0.2 m/s^2 in period 1. However, as we stated in the caption of Fig. 3, for those seismograms recorded in period 1, we only used velocities computed on time windows having peak acceleration lower than 0.2 m/s^2 in the moving-window analysis.

4. P6 “As of March 2019 ~ permanent velocity change”: I think this is an overstatement as no stations show an excess recovery over the preseismic level beyond the confidence interval at 8 yr in Fig. 3a.

Response: Thank you for your comments. We did not notice the confidence interval of the preseismic level in the original manuscript. According to your comments, we have removed the second sentence from the manuscript as shown below.

“As of March 2019, the recovery of the shear-wave velocity did not seem to have stabilized, even for three of the stations for which the shear-wave velocities had recovered to their pre-seismic level. ~~This is likely because irreversible changes (e.g., the consolidation of surface sediments) over the eight years after the earthquake would lead to a permanent velocity change.~~ For those stations whose near-surface seismic velocities had not yet recovered to the pre-seismic level, the time required to return to the pre-seismic level at the current recovery rate of the second stage ranged from several months to two hundreds years with a median of 15 years.”

5. P6 “We also noticed that ~ largest relative velocity changes”: Are there any evidences of cyclic mobility in waveforms of IBRH11?

Response: Thank you for your comments. According to your comments, we rechecked the waveforms of IBRH11, and found some evidences to support the occurrence of cyclic mobility at this station, as shown in Figure S6.

Figure S6: (a) Horizontal acceleration seismograms of the 2011 Tohoku-Oki Earthquake at IBRH11; (b) Enlarged part of the seismograms marked with a red box in the left subplot. The abrupt drop in the waveform amplitude before and after the last significant acceleration peak of the surface seismograms indicates the possibility of incipient quasi-liquefaction at this station.

6. The time scale of the first recovery stage (about 200 s) is comparable to the duration of the strong ground motion of the Tohoku-Oki Earthquake (Fig. 2a). I think this indicates that the first recovery merely reflects the classical nonlinear elasticity: as the amplitude of coda wave attenuates, the shear modulus recovers. To confirm this hypothesis, I recommend the authors to make a plot of the ground motion level versus the velocity reduction ratio for the mainshock seismogram, which shows the amplitude-dependence of the velocity (classical nonlinear elasticity). If the plotted curve behaves like a multivalued function (more than two values for a given ground motion level), the difference between the two velocity reduction values may be attributed to the non-classical nonlinearity.

Response: Thank you for your comments. These comments really help us a lot. We think this is a very good point to inspire us to further understand the nonlinear behavior of near-surface geomaterials during strong ground motions. According to your suggestions, we have made a plot of the shear modulus degradation versus the dynamic strain for the mainshock seismograms, as shown in Figure R2. It can be found that the shear modulus decreases with the increase of the dynamic shear strain before the recovery begins. Then, during the first recovery stage, the shear modulus recovers as the dynamic shear strain increases. However, the recovery does not follow the original descending path but is associated with hysteresis, which may be attributed to the non-classical nonlinearity. Besides, Bonilla et al. (2019) also reported similar findings, indicating the universality of non-classical nonlinearity in geotechnical materials during strong ground motions.

Figure R2. Shear modulus degradation as a function of dynamic shear strain for the mainshock seismograms. Note that the shear modulus is proportional to the square of velocity $G/G_0 \approx (V_s/V_{s0})^2$. The dynamic shear strain is calculated by $v(t)/Vs30^*$, in which $v(t)$ is the maximum value in each second of the velocity time histories at the surface, and $Vs30^*$ is the corrected $Vs30$. According to Bonilla et al. (2019), $Vs30^* = Vs30 \times V_s/V_{s0}$.

Reference

Bonilla, L. F. et al. Monitoring coseismic temporal changes of shallow material during strong ground motion with interferometry and autocorrelation. *Bulletin of the Seismological Society of America* **109**, 187-198 (2019).

<https://doi.org/10.1785/0120180092>

7. P7 “This may be ~ requires further study”: I think scale of the target material is an important factor that determines duration of the second recovery. The scale of soil layer ($\sim 10^2$ m) is much larger than the sample of Berea sandstone ($\sim 10^{-1}$ m), which may result in longer recovery time for the former.

Response: Thank you for your comments. According to the comments from you and another reviewer, we have added ‘the scale of the target material’ and ‘dynamic strain amplitude’ as two possible factors to determine the duration of the second recovery, as listed below.

“The slope of the second stage in the field observations is much lower than that in the laboratory experiments, leading to a significantly longer second recovery stage for the case of the Tohoku-Oki Earthquake. This may be related to the scale of the target material, dynamic strain amplitude, excitation time, confining pressure and geomaterial type. More experimental data and a broader quantitative comparison between laboratory experiment and field observation will be needed to further validate the present theoretical model and establish its limitations.”

8. Fig.3b,c: For the field observation (Fig. 3b), the two stage log-linear recovery is clearly observed. However, for the Shokouhi et al. (2017)’s experiment (Fig. 3c), the recovery curve does not show a clear bi-linear function if the red regression line is removed (see the inserted panel).

Response: Thank you for your comments. As you have already noticed, the results of some recent

experiments (e.g., Shokouhi et al., 2017) show deviations from a purely logarithmic law of recovery rate at early times of the recovery process. This phenomenon has been well explained by the theory proposed by Ostrovsky et al. (2019), as shown in Equations 10-12.

$$Dc/c = E_s/2E_0 = \frac{L}{2E_0} \ln \left[\frac{t}{\tau} + \left(1 - \frac{t}{\tau}\right) \exp(-|E_A/L|) \right] \quad (10)$$

According to Equation 10, the recovery is not necessarily logarithmic. In the beginning of the recovery, for small $t \ll \tau \exp(-|E_A/L|)$, we have

$$\Delta c/c = -|E_A|/2E_0 = \text{constant} \quad (11)$$

and then, due to $E_A/L \ll 1$, there exists a long-term logarithmic recovery:

$$\Delta c/c = \frac{\Lambda}{2E_0} \ln \left(\frac{t}{\tau} \right) \quad (12)$$

Moreover, as discussed in Ostrovsky et al. (2019), the deviations from the logarithmic law mainly depend on the value of $\Delta c(t=0)/c$. The higher the value of $\Delta c(t=0)/c$, the smaller the deviations. Note that the value of $\Delta c(t=0)/c$ is about 10^{-1} and 10^{-6} for the field observation and the Shokouhi et al. (2017)'s experiment, respectively. This can explain why the log-linear recovery can be observed more clearly for the field observation than for the Shokouhi et al. (2017)'s experiment.

9. P8 “According to the existing ~ stress-strain hysteresis”: If my understanding is correct, Sens-Schonfelder et al (2019, <https://academic.oup.com/gji/article/216/1/319/5116168>) proposed a model to explain the nonlinear elasticity based on the similar concept.

Response: Thank you for your comments. According to your comments, we have added this article as a reference to our manuscript as listed below.

“According to the existing adhesive contact theory (a more detailed description can be found in Lebedev and Ostrovsky, 2014 and Sens-Schönfelde et al., 2019), if the inter-grain distance exceeds a certain threshold under the action of external forces, the adhesive contact will be broken, resulting in a significant decrease in material modulus; and then the broken contact can be recovered at a closer inter-grain distance during unloading, leading to stress-strain hysteresis.”

10. The authors argue that the first recovery stage corresponds to the potential jump from the metastable contact #2 to #1. I think this idea is not appropriate to explain the results from the KiK-net study because of the reasons commented in #6. For the Shokouhi et al. (2017)'s experiment and the second recovery of the KiK-net results, the proposed idea (the potential jump from the metastable contact #1 to the stable contact) sounds reasonable.

Response: Thank you for your comments. These comments really help us a lot. An important evidence for the occurrence of the jump from the metastable contact #2 to #1 is that when the main shock ended, the seismic velocity had not yet returned to the pre-seismic level for all the selected stations, while the velocity reduction should be equal to zero when the main shock ended according to classic geotechnical laboratory experiments. On the other hand, the most significant difference between fast dynamics and slow dynamics is whether the external excitation ends. From this point of view, the first recovery is not entirely a strictly slow-dynamics process, because the seismic wave has not yet ended when the first recovery begins. In this case, the fast dynamics and slow dynamics can occur simultaneously. In a recent study, Ostrovsky et al. (2018) developed a modified version of the Arrhenius theory to simultaneously describe nonlinear fast and slow dynamics under sinusoidal dynamic wave excitation. In their theory, the variation of sound velocity in a material is presented as a

superposition of an oscillating and a slowly varying, period-averaged part, in which the former and the latter describe fast dynamics and slow dynamics, respectively. Nevertheless, the amplitude of the sinusoidal dynamic wave excitation considered in their theory and used in the corresponding experiment does not vary with time. It is still an open question how the velocity varies under variable amplitude dynamic wave excitation in both theory and lab experiment. At last, according to your comments and what we mentioned above, we have added a discussion on this issue to the Conclusion section, as listed below.

“However, many details of the physical mechanism remain unclear. For example, the first recovery stage in the field observations is not entirely a strictly slow-dynamics process, because the seismic wave has not yet ended when the first recovery begins. In this case, the fast dynamics and slow dynamics can occur simultaneously. In a recent study, Ostrovsky et al. (2018) developed a modified version of the Arrhenius theory to simultaneously describe nonlinear fast and slow dynamics under sinusoidal dynamic wave excitation. Nevertheless, the amplitude of the sinusoidal dynamic wave excitation considered in their theory and used in the experiment does not vary with time. It is still an open question how the velocity varies under variable amplitude dynamic wave excitation in both theory and lab experiment. We believe that the softening and healing of geomaterials should not be limited to the scope of seismology and geophysics but involves multiple disciplines, and further studies are needed in the future.”

Reference

Ostrovsky, L. et al. Nonlinear relaxation in geomaterials: New results. *Proceedings of Meetings on Acoustics* **34**, 032002 (2018).

<https://doi.org/10.1121/2.0000910>

11. Conclusions: Conclusion should focus on what authors did in this study, not the study of other researchers. The sentence of “For example ~ Vucetic and Dobry (1991)” is not suitable for the conclusion.

Response: Thank you for your comments. According to your comments, we have moved this sentence from the conclusion to a suitable place in the mainbody as shown below.

“In a recent study, it was found that the mean effective confining pressure is likely to be similar across different KiK-net stations, hence the degree of the near-surface velocity reduction is primarily affected by the shear strain and soil type, which was empirically estimated using the peak ground acceleration, the initial unperturbed near-surface shear-wave velocity, and the plastic index of the near-surface sediments (Wang et al., 2019). For example, according to the one-dimensional wave propagation theory in Beresnev and Wen (1996), a 5-Hz sinusoidal wave with an amplitude of 1 m/s^2 leads to a dynamic shear strain of 1×10^{-4} at a typical stiff-soil site of a shear-wave velocity of 320 m/s, which generates a shear modulus reduction of approximately 5-30 % depending on soil plasticity based on the data from geotechnical laboratory experiments in Vucetic and Dobry (1991).”

12. Station and record selection: I think the finally selected 12 stations are also influenced by strong earthquakes. Seismicity of the target region activated greatly after the Tohoku-Oki Earthquake, which includes Mj7.0 event occurred on April 11, 2011 (see <https://agupubs.onlinelibrary.wiley.com/doi/full/10.1029/2012GL051491>). Therefore, I’m concerned about contamination of another velocity reduction and recovery due to these large earthquakes.

Response: Thank you for your comments. These comments help us a lot. According to your comments,

we have carefully read the paper of Imanishi et al. (2012). Imanishi et al. (2012) reported an unusual shallow normal-faulting earthquake sequence occurred near the Pacific coast at the Ibaraki-Fukushima prefectural border after the occurrence of the Tohoku-Oki earthquake. This earthquake sequence contained an M7.0 event on 11 April 2011 (focal depth = 6 km) together with 17 moderate-magnitude shallow-focus events ($5 \leq M \leq 6.4$ & focal depth ≤ 15 km) as of 31 December 2011. It can be found that the source area of the earthquake sequence is quite limited and does not include any selected stations. For the 17 moderate-magnitude events, the PGAs of the seismograms recorded at the selected stations are rarely greater than 1 m/s^2 , while those for the M7 event range between 1.05 and 3.82 m/s^2 . Hence, we applied short-time moving-window seismic interferometry to the seismograms recorded at the 12 selected stations during the M7 event to investigate the coseismic velocity changes, as shown in Figure R4. For the selected stations, the maximum velocity changes during the M7 event range from 4.8% to 12.5%, which are far less than those during the Tohoku-Oki earthquake. It also should be noted that the velocity tends to be stable at the end of the seismograms for most of the selected stations. In a recent study, Qin et al. (2020) reported a velocity reduction of 4.5% in the upper 150 m at the Garner Valley Downhole Array (GVDA) under a moderate-level of ground motion ($\text{PGA} = 0.4 \text{ m/s}^2$) and a rapid full recovery within about 240 s. Considering the relatively high PGAs at a few stations during the 11 April 2011 M7 event, we acknowledge that the 11 April 2011 M7 event may cause a limited effect on shear wave velocity for several days or even months for those stations, which will be part of our future work. As far as we know, a velocity model for the combination of two recovery processes has been proposed in Vidale and Li (2003), as shown in Figure R5. Finally, we would like to emphasize that the recovery process of the 11 April 2011 M7 event can hardly be observed in Figure 3a due to the logarithmic time scale used, which also means that it has little influence on our results.

Figure R3. Distribution of the selected stations and the source area of the normal-faulting earthquake sequence (shaded area, modified from Imanishi et al., 2012). The red circle represent the epicenter of the 11 April 2011 M7 event.

Figure R4. Horizontal seismicograms of the 11 April 2011 M7 event at the selected KiK-net stations and corresponding coseismic variations of near-surface shear-wave velocity. The horizontal seismicograms are obtained by rotating the north-south (NS) and east-west (EW) components from 10° to 180° in 10° increments, in which the azimuth is measured clockwise from the north. Here shows the horizontal seismicograms with the maximum PGA among the eighteen directions. The maximum PGA is given by the tick label of the acceleration-axis, and the corresponding azimuth (θ) is listed in each subplot. The upper and lower tick label of the velocity-axis represent the pre-seismic level of the seismic velocity before the Tohoku-Oki Earthquake and the lowest velocity during Tohoku-Oki Earthquake, respectively. The maximum velocity change during the 11 April 2011 M7 event (by percent) is listed in the bottom-right corner of each subplot.

Figure R5 Model of velocity as a function of time owing to damage from two earthquakes and their combination compared with observations (Vidale and Li, 2003). Shown is healing as a logarithm of time, although details just after each event and extrapolating into the future are not well constrained. The velocity before the Landers earthquake was not measured.

Reference

Imanishi, K. et al. Unusual shallow normal-faulting earthquake sequence in compressional northeast Japan activated after the 2011 off the Pacific coast of Tohoku earthquake. *Geophysical Research Letters* **39**, L09306 (2012).

<https://doi.org/10.1029/2012GL051491>

Qin, L. et al. Imaging and monitoring temporal changes of shallow seismic velocities at the Garner Valley near Anza, California, following the M7.2 2010 El Mayor-Cucapah earthquake. *Journal of Geophysical Research: Solid Earth* **125**, e2019JB018070 (2020).

<https://doi.org/10.1785/0120180092>

Vidale, J. E, Li, Y. G. Damage to the shallow Landers fault from the nearby Hector Mine earthquake. *Nature* **421**, 524–526 (2003).

<https://doi.org/10.1038/nature01354>

13. Station and record selection: In the selection of analyzed seismograms, S/N ratio should be much larger than one within the target frequency range because noise is mostly composed of surface wave and assumption of vertical incidence (must be satisfied in the deconvolution interferometry) is not appropriate for noise.

Response: Thank you for your comments. We had recognized that the deconvolution interferometry is not appropriate for noise. Hence, in our manuscript, as we mentioned in the Method section, the first step in using deconvolution interferometry is to remove the pre-event noise from the seismograms after picking the P wave arrival.

Replies to Reviewer #2

Comments:

The authors investigated the near-surface material softening process after the 2011 Tohoku earthquake using seismic interferometry. The seismic interferometry uses co-located seismic stations at different depths from the KiK-net. The differential travel time for a given earthquake between the two stations at the same site is used to estimate the local shear-wave velocity at the near-surface. This is a standard technique and has been proven useful to monitor near-surface velocity changes after large earthquakes. The authors worked on 3762 earthquakes to obtain an eight-year time series of the velocity reduction-recovery process in the Honshu region after the Tohoku earthquake. They found a clear velocity reduction at all sites due to the Tohoku earthquake that occurred during the passage of the seismic waves and a two-stage recovery process that is still ongoing to date. The first stage healed about 60% of the velocity reduction within hundreds of seconds and the second stage may take up to 15 years to fully recover to the previous local strength. To understand the process, the authors compared the observations to a laboratory experiment and analytical models. The laboratory experiment that the author cited also demonstrates a two-stage recovery process that is similar to the field observations. With the assumed model setup, which includes two solid surfaces sandwiching a water layer, the authors obtained solutions that are consistent with the field observations.

It is a very interesting paper and I enjoyed reading it. I am not an expert of the inter-grain interaction modeling, so I will defer it to other reviewers' expert opinions. The seismic data processing is rigorous and well-reasoned. In particular, I appreciate the author systematically examined the shear-wave velocity recovery process over eight years. However, the comparison between the laboratory experiment and the field observations requires more justifications. The laboratory experiment investigated the P-wave velocity evolution process, and it is unlikely to be the same as that of S-waves. Besides, the shear strains of the field observations are two to three orders larger than the laboratory environment. Assuming the recovering process depends on the state and the rate as the analytical modeling suggests, this large deviation may lead to different processes. It is always challenging to scale lab results to field observations, and the apparent agreement between the experiment and the observations may not be consistent.

Response: We really appreciate your detailed comments and suggestions. We have tried our best to revise the manuscript according to your comments. In the annotated version of the revised manuscript, all changes are highlighted in red. Thank you again for your careful review of our manuscript.

As you mentioned, special attention is required for the comparison between the laboratory experiment and the field observations. The first concern is the differences between the recovery of P-wave and that of S-wave, which has been discussed in detail by Averbakh et al. (2017) and Ostrovsky et al. (2019). The observations reported by Averbakh et al. (2017) demonstrated that the recovery of both bulk and shear moduli for carbonate rock samples are time logarithmic. Ostrovsky et al. (2019) suggested that both shear and tensile deformations are able to break contacts leaving their small fraction in a metastable state. Moreover, according to the well-known Mindlin's theory, shear and tensile deformations are interrelated. Note that the Poisson's ratio of geomaterials is usually considered to be constant below a strain level of about 10^{-4} ~ 10^{-5} (Dutta and Saride, 2015; Chen et al., 2018); in this case, both compressional and shear wave velocities are proportional to each other.

However, as we emphasized in our manuscript, a notable difference between the normalized P-wave and S-wave velocity recovery processes is the recovery rate in the second stage. According to

the comments from you and another reviewer, we have added ‘the scale of the target material’ and ‘dynamic strain amplitude’ as two possible factors to determine the duration of the second recovery, as listed below. Besides, it should be pointed out that more laboratory experimental data and a broader quantitative comparison between laboratory experiment and field observation will be needed to further validate the present theoretical model and establish its limitations.

“The slope of the second stage in the field observations is much lower than that in the laboratory experiments, leading to a significantly longer second recovery stage for the case of the Tohoku-Oki Earthquake. This may be related to **the scale of the target material, dynamic strain amplitude**, excitation time, confining pressure and geomaterial type. **More experimental data and a broader quantitative comparison between laboratory experiment and field observation will be needed to further validate the present theoretical model and establish its limitations.**”

Reference

Dutta, T.T., Saride, S. Influence of Shear Strain on the Poisson’s Ratio of Clean Sands. *Geotechnical and Geological Engineering* **34**, 1359-1373 (2016).

<https://doi.org/10.1007/s10706-016-0047-1>

Chen et al. An effective way to estimate the Poisson's ratio of silty clay in seasonal frozen regions. *Cold Regions Science and Technology* **154**, 74-84 (2018).

<https://doi.org/10.1016/j.coldregions.2018.06.003>

Averbakh, V.S., Bredikhin, V.V., Lebedev, A.V. et al. Nonlinear acoustic spectroscopy of carbonate rocks. *Acoustical Physics* **63**, 346-358 (2017).

<https://doi.org/10.1134/S1063771017030022>

Replies to Reviewer #3

General comments

I have reviewed the paper entitled “Near-surface softening and healing in eastern Honshu associated with the 2011 magnitude-9 Tohoku-Oki Earthquake” by Su-Yang Wang, Hai-Yang Zhuang, Hao Zhang, Hong-Jun He, Wei-Ping Jiang, Er-Lei Yao, Bin Ruan, Yong-Xin Wu and Yu Miao. The paper deals with interesting co-seismic phenomena that involve material softening and subsequent healing. This behavior has already been observed for a long time in rock and concrete, but it is relatively new in soils. The authors propose some theoretical hypotheses to explain such observations in records from the 2011 M9 Tohoku earthquake. The paper is in general well written and clear. However, I have some comments that I hope will help to improve this interesting work. In any case, I support the publication of this article after the corrections are made.

Response: We really appreciate your detailed comments and suggestions. We have tried our best to revise the manuscript according to your comments. In the annotated version of the revised manuscript, all changes are highlighted in red. Thank you again for your careful review of our manuscript. Itemized response to your questions is appended below.

Specific comments & Point-by-point response

This paper uses records from the 2011 M9 Tohoku earthquake to study the nonlinear behavior of shallow material, namely the softening and healing processes during strong shaking. The authors use seismic interferometry to estimate the velocity changes from surface and borehole records. This technique is well described in Nakata and Snieder (2012) and Bonilla et al. (2019), yet this technique is mentioned without telling what it measures (i.e., travel time), and since it is the base for the discussion, I think it deserved more explanation in the text body or say that the details will be seen in the supplementary material. For example, time window length, the particular formulation of the seismic interferometry (you use multitaper deconvolution), overlapping between windows. And a very important information that is missing is how the velocity change is computed? What components do you use (NS, EW, UD, a combination of them)? What is the frequency band that you study at each station, and how do you estimate it? As it is written, it seems that this operation is trivial, and it is not and this has been shown in the references above. In other words, you need to convince the readers that your computations are stable before analyzing their results.

Response: Thank you for your comments. According to the journal’s format requirements (see in www.nature.com/ncomms/submit/how-to-submit), the Methods section is placed after the main text of the manuscript and before the References. The Methods section contains all the technical information your mentioned above.

Another issue is the earthquakes used in this study. The time span is at least 8-9 years after the 2011 mainshock. Are the events coming from the same epicentral area or they come from all over Japan? Is there any depth constraint? Do you use the same events for all stations? or you have a set of earthquakes for each station. Once this is done, how do you estimate the error bars in the velocity changes? Qin et al. (2020) compute the error bars from the statistics of the travel times when using either seismic interferometry or autocorrelation functions.

Response: Thank you for your comments. The deconvolution-based interferometry has proven to effectively eliminate the effects of the incident wavefields for vertical borehole arrays [see “Appendix

A: 1-D Seismic Interferometry” in Nakata and Snieder (2012)]. Hence, source location and magnitude are not used as the screening parameters for record selection in our manuscript. As shown in Figure S4, most of the epicenters of the selected events are located in Northeast Honshu and surrounding waters. In addition, to ensure that the seismic waves propagate in a near-vertical direction between the borehole and surface seismograph, we selected seismograms with focal depths of at least 7 km (Nakata and Snieder, 2011). A detailed introduction to the seismic data used in our study can be found in the “Station and record selection” of the Method section.

As mentioned in the Method section, we rotated the NS and EW components of the seismograms from 10° to 180° in 10° increments to synthesize horizontal seismograms in eighteen directions, so the near-surface seismic velocity can be expressed as a function of azimuth angle by dividing the borehole depth by the wave propagation time in different directions. Then the isotropic term was chosen to represent the near-surface velocity, in which the isotropic term is calculated as the average velocity in the eighteen directions. In our manuscript, the error bars were computed directly from the statistics of the seismic velocities (rather than the travel time) measured from different events over a time span (see the caption of Figure 3).

Figure S4 (a) Distribution of the epicenters of the selected earthquakes; (b) Magnitude and focal depth of the selected earthquakes; (c) PGA and epicentral distance of the selected records.

There is no mention on how the mean velocity (and the error estimate) is computed at each station. Given the numbers, I suppose it is the average shear wave speed for the whole soil column. The problem that you might have with geotechnical people is that the numbers are relatively large, and a strong velocity change is reported. Thus, in their point of view, how a material that is close to 900 m/s can change up to 20%. Since I worked at station IBRH16 (Bonilla et al., 2019), and due to the use of autocorrelation functions and the knowledge of the velocity profile, it is probably the first tens of meters that produced such velocity reduction because deeper layers have larger shear velocities. In other words, the nonlinear behavior is rather shallow and localized in the softer soils close to the

surface. I recommend to write something about this because the theory that is used is for granular material, which is more likely to happen in the shallow part only. I would probably plot the velocity profiles to discuss this issue.

Response: Thank you for your comments. These comments really help us a lot. In our manuscript, as mentioned in the Method section, the near-surface velocity is computed by dividing the borehole depth by the travel time from borehole bottom to ground surface. Hence, the near-surface velocity indicates the average velocity for the whole soil column. We have carefully read your recent paper of Qin et al. (2020), which shows that almost all the velocity reduction is localized in shallow layer up to 22 m at the Garner Valley Downhole Array (GVDA) under a moderate-level of ground motion ($PGA = 0.4 \text{ m/s}^2$) caused by the 2010 M7.2 El Mayor-Cucapah (EMC) earthquake. We also agree that the shallow soft layers exhibit stronger nonlinearity under strong ground motion than the deep hard layers. According to your suggestions, we have plotted the shear velocity profiles of those selected stations, as shown in Figure S7. It can be found that the shallow layers have much lower velocities than the deep layers. For the selected stations, the velocities of the topmost layer, the average velocities to 30 m, and the average velocities to borehole bottom are 60-240 m/s, 230-590 m/s, 400-2300 m/s, respectively. In addition, the elastic nonlinearity of granular materials decreases with increasing confining pressure. Hence, it is probably that the shallow soft sediments of the top tens of meters play a major role in the near-surface softening, and a greater velocity reduction could be expected in the shallow subsurface compared to the average velocity reduction from the ground surface to the borehole bottom. Accordingly, we have added a discussion on the vertical distribution of the velocity reduction to the Conclusion section, as listed below.

“Elastic wave velocity is an indicator of modulus, hence our observation demonstrates that a great earthquake can cause significant softening and long-term healing in the near surface over a wide area (Figs. 2, 3a). For those 12 selected stations, the maximum coseismic velocity reduction in the upper 100-200 m during the 2011 Mw9 Tohoku-Oki earthquake ranged from 11.9% to 46.3%. Note that the velocity reduction is unevenly distributed in vertical direction, and it is probably that most of the velocity reduction is localized in shallow layers (Bonilla et al., 2019; Qin et al., 2020). This is because the shallow layers have much lower velocities than the deep layers (Fig. S7), and the elastic nonlinearity of granular materials decreases with increasing confining pressure (Johnson and Jia, 2005). Hence, the shallow soft sediments of the top tens of meters play a major role in the near-surface softening, and a greater velocity reduction can be expected in the shallow subsurface compared to the average velocity reduction from the ground surface to the borehole bottom.”

Figure S7. Shear wave velocity profiles of the selected seismic stations from P-S logging tests (the profile data are available from https://www.kyoshin.bosai.go.jp/kyoshin/db/index_en.html).

Regarding the hypothesis that pore pressure and in particular, dilatancy effects might happen, I am not sure how you can demonstrate this. I also recommend reading our paper Qin et al. (2020) where we studied a borehole array with pore pressure transducers, and it was difficult to relate pore pressure effects and velocity changes. In my opinion, the grain arrangement during the shaking is more important than pore pressure effects if the material is not fully saturated. However, Tohoku produced such large PGA's that both effects could be present at the same time. This is not the case with our study in California. In any case, this needs more justification, or just soften the text because it is a not easy subject. The method that you describe Kostadinov and Yamazaki (2001) basically computes the instantaneous frequency, which is also shown in Bonilla et al. (2019), but no evidence of liquefaction at the site was found. Thus, again, this is a tricky question, and the change of frequency can also be related to the change of the velocity change itself, thus it is a circular thinking.

Response: Thank you for your comments. These comments help us a lot. As mentioned in Régnier et al. (2016), from a mesoscopic point of view, nonlinear soil behavior involves rather complex mechanical processes, which may be grouped roughly in two main classes as follows: (1) the variation of the mechanical properties of the material, which is usually characterized by a degradation in the shear modulus coupled with an increase in the anelastic damping; (2) the increase of pore water pressure of saturated granular soils, related to volumetric changes of the soil skeleton under shear stress, which may cause liquefaction if the pore pressure exceeds the contact stress among the soil grains.

After reading your recent paper of Qin et al. (2020), we agree that for the stations selected in our manuscript, both processes could be involved during the Tohoku-Oki earthquake. We also applied the liquefaction detection methods described in Kostadinov and Yamazaki (2001) to the seismograms used in Bonilla et al. (2019), and no liquefaction was detected by those methods. Of course, the liquefaction detection methods described in Kostadinov and Yamazaki (2001) are empirical, and the results may be misjudged. Hence, according to your comments, we have preferred to soften the text, as show below.

“The coseismic velocity variations at FKSH14 indicate ~~highly~~ suspected transient liquefaction in the shallow surface layer, which can be confirmed by two existing liquefaction detection methods using ground motion records from Kostadinov and Yamazaki (2000) and is also visually evidenced by the abrupt drop in the waveform amplitude before and after the last significant acceleration peak of the surface seismograms (Fig. S5). Hence, considering a solidification process after **the suspected transient** liquefaction ~~accompanied by dissipation of pore water pressure~~ (Wang et al., 2013), we defined the start time of the healing process as the time with the fastest velocity recovery rate for FKSH14, and this definition is also applicable to almost all the stations.”

Regarding the theory you use to explain the observations, I suggest to give a proper credit to the latest work of Ostrovsky et al. (2019). Indeed, you basically use their technique on the results of your computations. I may be missing something, but the new thing here is the application of such technique in Tohoku data, and it works quite well. Yet again, I am worry about the geotechnical community because you report velocity changes for the whole soil column and the processes can be very different at each soil condition present in the column. What is interesting for me is that there is a trending behavior, and probably all materials have it, what we see is the average, and you could write like that to avoid critics on a particular nonlinear soil behavior.

Response: Thank you for your comments. In our study, the framework of the theory used to explain the observations generally follows the latest work of Ostrovsky et al. (2019). Besides, based on their work on the double-well potential energy model, we proposed a multiple-well potential energy model that includes a concurrence between the Lennard-Jones potential and hydration interaction, which can be used to explain the emergence of different stages in the healing process. Therefore, according to your comments, we have revised the relevant sentences in the Conclusion section to highlight our technical improvements and the importance of the work of Ostrovsky et al. (2019), as listed below. Meanwhile, as mentioned above, we have added a discussion on the vertical distribution of the velocity reduction to the Conclusion section to avoid critics on a particular nonlinear soil behavior.

“Finally, **based on the work of Ostrovsky et al. (2019)**, we propose a theoretical **multiple-well** model of the contact force potential between grains in geomaterials, and describe the slow relaxation as the transition process from **a metastable contact to another metastable contact or the stable contact** (Fig. 4).”

Another interesting consequence is the recovery time that takes almost 10 years for these stations. This is something that is not taken into account in traditional seismic hazard. This means, that even if we characterize well a site, the known site response is not going to reduce the ground motion variability. If a strong perturbation takes place, the system will take time to recover, and this is proportional to the amplitude of the perturbation. This is critical for site-specific seismic hazard estimates.

Response: Thank you for your comments. These comments help us a lot. According to your comments, we have added two sentences to the Conclusion section to illustrate the potential impact of the

long-term recovery process on seismic hazard estimation, as listed below.

“ Then we find that about two-thirds of the initial reduction is recovered rapidly in a few minutes, and the remaining one-third is recovered slowly over months or years (Figs. 3b, 3c). It also should be noted that the long-term recovery process has not been taken into account in traditional seismic hazard yet. These findings indicate that in site-specific seismic hazard estimates, it may be necessary to reevaluate the seismic site response after a large earthquake occurs to consider the effects of long-term changes in near-surface stiffness caused by strong perturbations.”

REVIEWER COMMENTS

Reviewer #1 (Remarks to the Author):

I read the author's response and the revised manuscript, and found my comments are mostly reflected in the new manuscript. However, I suggest the author to consider to include Figure R2, the shear modulus reduction ratio with respect to the dynamic strain, in the manuscript. This figure clearly shows contribution of fast and slow dynamics during strong motion. The residual between the red and blue points corresponds to the contribution of slow dynamics (non-classical nonlinearity), and this residual should be focused on when discussing the first recovery stage. As the velocity change due to slow dynamics only is smaller than the whole velocity change due to both slow and fast dynamics, the slope of the first recovery stage (Fig. 3) will be gentler if only the contribution of slow dynamics is plotted. I recommend the author to revise Figure 3 as I suggested and confirm whether the two recovery stages are still visible even after this correction. This makes this paper much more powerful and compelling.

Another minor comment is to refer following DOI number when citing KiK-net.

DOI:10.17598/nied.0004

Please read (1) of

https://www.kyoshin.bosai.go.jp/kyoshin/docs/overview_kyoshin_index_en.html
for reference of the KiK-net data.

Reviewer #2 (Remarks to the Author):

I have read the revised manuscript and the reply to the reviews. The revision has addressed most of the comments from both reviewers. I recommend the paper for publication.

Reviewer #3 (Remarks to the Author):

Thanks for taking into account my suggestions. The paper is fine for me. I only have a small additional suggestion. I still think that showing Figure 2 as it is, shows the hypothesis that the velocity reduction is for the whole depth of the borehole station. You have added a comment related to the fact that these reductions are probably related to the shallow layers that are shown in Figure S7. To avoid any misinterpretation, why you do not plot the velocity reduction as a function of time, instead of the product of average velocity with velocity reduction? In this way you do not claim where in the borehole this is happening, which needs further studies.

Sincerely,

Fabian Bonilla

Replies to Reviewer #1

Comments:

I read the author's response and the revised manuscript, and found my comments are mostly reflected in the new manuscript. However, I suggest the author to consider to include Figure R2, the shear modulus reduction ratio with respect to the dynamic strain, in the manuscript. This figure clearly shows contribution of fast and slow dynamics during strong motion. The residual between the red and blue points corresponds to the contribution of slow dynamics (non-classical nonlinearity), and this residual should be focused on when discussing the first recovery stage. As the velocity change due to slow dynamics only is smaller than the whole velocity change due to both slow and fast dynamics, the slope of the first recovery stage (Fig. 3) will be gentler if only the contribution of slow dynamics is plotted. I recommend the author to revise Figure 3 as I suggested and confirm whether the two recovery stages are still visible even after this correction. This makes this paper much more powerful and compelling.

Another minor comment is to refer following DOI number when citing KiK-net. DOI:10.17598/nied.0004. Please read (1) of https://www.kyoshin.bosai.go.jp/kyoshin/docs/overview_kyoshin_index_en.html for reference of the KiK-net data.

Response: We really appreciate your constructive comments and suggestions. According to your suggestions, we have added Figure S8 and a citation of NIED (2019) to our revised manuscript, as shown below.

“However, many details of the physical mechanism remain unclear. For example, the first recovery stage in the field observations is not entirely a strictly slow-dynamics process, because the seismic wave has not yet ended when the first recovery begins. In this case, the fast dynamics and slow dynamics can occur simultaneously (Figure S8).”

Fig. S8. Shear modulus degradation as a function of dynamic shear strain for the mainshock seismograms. The shear modulus is proportional to the square of velocity $G/G_0 \approx (V_s/V_{s0})^2$. The dynamic shear strain is calculated by $v(t)/V_{s30}^*$, in which $v(t)$ is the maximum value in each second of the velocity time histories at the surface, and $V_{s30}^* = V_{s30} \times V_s/V_{s0}$. During the recovery process, there is a certain correlation between the shear modulus ratio and the dynamic strain for a majority of the stations,

but the recovery of the shear modulus does not follow the original descending path, which is attributed to the combined effect of slow and fast dynamics.

“Seismic data used in the study are provided by National Research Institute for Earth Science and Disaster Resilience (NIED, 2019) and are available at: <http://www.kyoshin.bosai.go.jp/>. Source data are provided with this paper.”

Furthermore, following your suggestions, we have tried to correct the recovery rate (slope) of the first stage. First, we used a classic hyperbolic model to fit the relationship between shear modulus degradation and dynamic shear strain before recovery, as shown in Figure R1. Then, we separated the contributions of fast and slow dynamics in shear modulus degradation during the first stage of the recovery process, as shown in Figure R2 and Equation R1. Finally, the corrected velocity change is calculated by Equation R2, as shown in Figure R3.

$$1-(V_s / V_{s0})^2 = \Delta_1 + \Delta_2 \quad R1$$

$$(\Delta c / c)_{cor} = \sqrt{1 - \Delta_2} - 1 \quad R2$$

Figure R1 Shear modulus degradation as a function of dynamic shear strain for the mainshock seismograms. The black line represents the fitted-curve of the red points.

Figure R2 A sketch of the contributions of fast and slow dynamics in shear modulus degradation during the first stage of the recovery process, in which Δ_1 and Δ_2 represent the contributions of fast and slow dynamics, respectively.

Figure R3. Comparison between the corrected and uncorrected velocity changes during the first stage of the recovery process for the mainshock seismograms.

From Figure R3, it can be found that for a majority of the stations, the corrected velocity shows a decreasing trend, rather than an increasing trend with a gentler slope. I think there are two possible reasons. First, the decrease of shear wave velocity before the recovery is not entirely a strictly fast dynamic process. It was found that the velocity reduction is related to the time of the dynamic wave excitation, as shown in Figure R4. Second, the classical nonlinearity and non-classical nonlinearity do not satisfy the simple superposition principle. Therefore, it is not feasible to decouple the slow and fast dynamics directly by subtraction. As we stated, it is still an open question how the velocity varies under variable amplitude dynamic wave excitation in both theory and lab experiment. In my opinion, both fast and slow dynamics play important roles under constant-amplitude dynamic wave excitation. For gradually decreasing dynamic wave excitation, slow dynamics plays a critical role, while for gradually increasing dynamic wave excitation, fast dynamics plays a critical role. We will leave it to a further study.

Thank you again for your careful review of our manuscript.

Figure R4. Left: Dynamics of relative change of sound velocity in a sample of Berea sandstone during and after the oscillating impact. Right: the corresponding theoretical modeling (Ostrovsky et al., 2018).

Reference

National Research Institute for Earth Science and Disaster Resilience. NIED K-NET, KiK-net. (2019)

<https://doi.org/10.17598/NIED.0004>

Ostrovsky, L. et al. Nonlinear relaxation in geomaterials: New results. *Proceedings of Meetings on Acoustics* **34**, 032002 (2018).

<https://doi.org/10.1121/2.0000910>

Comments

Thanks for taking into account my suggestions. The paper is fine for me. I only have a small additional suggestion. I still think that showing Figure 2 as it is, shows the hypothesis that the velocity reduction is for the whole depth of the borehole station. You have added a comment related to the fact that these reductions are probably related to the shallow layers that are shown in Figure S7. To avoid any misinterpretation, why you do not plot the velocity reduction as a function of time, instead of the product of average velocity with velocity reduction? In this way you do not claim where in the borehole this is happening, which needs further studies.

Response: We really appreciate all your comments and suggestions. According to your suggestions, we have revised Figures 2 and 3 to plot the velocity reduction as a function of time, instead of the product of average velocity with velocity reduction, as shown below. Thank you again for your careful review of our manuscript.

Fig. 2: Horizontal seismograms of the Tohoku-Oki Earthquake at the selected KiK-net stations and corresponding coseismic variations of near-surface shear-wave velocity. The horizontal seismograms are obtained by rotating the north-south (NS) and east-west (EW) components from 10° to 180° in 10° increments, in which the azimuth is measured clockwise from the north. Here shows the horizontal seismograms with the maximum PGA among the eighteen directions. The maximum PGA is given by the tick label of the acceleration-axis, and the corresponding azimuth (θ) is listed in each subplot. **The lower tick label of the velocity-change-axis represents the maximum velocity reduction (by percent) during the mainshock compared with the preseismic**

level of the seismic velocity before the Tohoku-Oki Earthquake. The red circle indicates the initiation of the recovery process.

Fig. 3: (a) Recovery process of near-surface shear-wave velocity for the selected KiK-net stations. The gray dots represent raw data computed on time windows having peak acceleration lower than 0.2 m/s^2 . The horizontal dashed line and gray-shaded band show the mean and standard deviations of the velocities before the Tohoku-Oki Earthquake. The black dots and the error bars indicate the mean and standard deviations for **velocity changes** measured over a time span of 0.5 in the logarithmic scale. The bi-linear red lines are the regression lines of the mean values. The fitting of the first recovery stage is obtained from the first five data points for each stations except for IBRH11, for which the first two points are ignored for fitting. The fitting of the second recovery stage is obtained from the rest of data points. The details regarding the regression are provided in Table S2. In each subplot, the red arrow on the left shows the abscissa of the intersection of the two regression lines, whereas the red arrow on the right shows the abscissa of the intersection between the regression line of the second stage and the preseismic velocity level. (b) Normalized shear-wave velocity recovery process from field observations. The red bi-linear lines and arrows are the same as in Fig. 3a. The black dotted line and gray-shaded band represent the averaged

normalized velocity recovery process and the corresponding 63% confidence limit, respectively. α_1 and α_2 are the log slopes in the first and the second stages, respectively; Δc_1 and Δc_2 are the recovery amounts of the shear-wave velocity during the first and the second stages, respectively. The relative velocity changes for different stations before normalization are shown in the bottom right corner. (c) Normalized compressional-wave velocity recovery process from laboratory experiments (modified from Shokouhi et al., 2017). The red arrow on the left indicates the initiation of the first stage of the recovery process.

REVIEWERS' COMMENTS

Reviewer #1 (Remarks to the Author):

I read the authors' reply and revised version and am satisfied with the revised work. I think this paper is ready for publication.

I also thank the authors for precise analysis trying to separate the fast and slow dynamics-related recoveries in the first recovery stage. I agree that classical and non-classical nonlinearities cannot be described by simple superposition: the detail of the mechanism should be investigated in the future study.